# Active Evaluation Acquisition for Efficient LLM Benchmarking

**Yang Li** [1]  **Jie Ma** [1]  **Miguel Ballesteros** [1]  **Yassine Benajiba** [1]  **Graham Horwood** [1]

## Abstract

As large language models (LLMs) become increasingly versatile, numerous large scale benchmarks have been developed to thoroughly assess their capabilities. These benchmarks typically consist of diverse datasets and prompts to evaluate different aspects of LLM performance. However, comprehensive evaluations on hundreds or thousands of prompts incur tremendous costs in terms of computation, money, and time. In this work, we investigate strategies to improve evaluation efficiency by selecting a subset of examples from each benchmark using a learned policy. Our approach models the dependencies across test examples, allowing accurate prediction of the evaluation outcomes for the remaining examples based on the outcomes of the selected ones. Consequently, we only need to acquire the actual evaluation outcomes for the selected subset. We rigorously explore various subset selection policies and introduce a novel RL-based policy that leverages the captured dependencies. Empirical results demonstrate that our approach significantly reduces the number of evaluation prompts required while maintaining accurate performance estimates compared to previous methods.

## 1. Introduction

As large language models (LLMs) become increasingly versatile, comprehensive benchmarks have emerged to assess their capabilities. However, these evaluations incur substantial costs - for example, evaluating on the HELM benchmark requires 4,200 GPU hours for a 176B BLOOM model and $9,337 for text-davinci-002 API calls (Liang et al., 2022). Such costs hamper development by preventing frequent evaluation during model training and extensive hyperparameter tuning during inference.

In this work, we aim to improve evaluation efficiency by reducing the number of evaluation prompts needed. We observe that evaluation prompts are often correlated, where a model's performance on certain prompts tends to correspond with its performance on related ones. To leverage this, we build a model to capture dependencies across prompts and predict evaluation scores based on observations from a subset of prompts. Our goal becomes identifying the minimal subset that can accurately recover the evaluation scores for the remaining prompts.

Instead of using a fixed subset across all models, we propose selecting unique subsets for each model through active evaluation acquisition (AEA). This approach recognizes that models may have varying strengths - for example, one model may excel in arithmetic reasoning while another shows stronger commonsense reasoning. By tailoring the subset of prompts for each model, we ensure a more accurate and targeted evaluation of its capabilities. Furthermore, our dynamic acquisition process adapts in real time as evaluation scores are gathered. As the model's performance on initial prompts is observed, the system adjusts subsequent prompt selections to better explore areas of uncertainty or confirm early findings. This iterative approach not only enhances the accuracy of performance estimation but also reduces redundancy by avoiding prompts that are likely to yield predictable results, thereby saving computational resources and time. Importantly, the final evaluation score is derived from both the acquired scores on selected prompts and predicted scores on the remaining prompts, ensuring comparability across models is maintained.

Our contributions include: 1) We tackle LLM evaluation efficiency through dependency modeling and subset selection, connecting it with subset selection literature. 2) We design a generative model that captures dependencies across prompts and handles mixed-type evaluation scores. 3) We thoroughly evaluate existing subset selection algorithms on several popular LLM benchmarks. 4) We develop new subset selection policies, with our RL-based approach achieving the best performance using minimal acquisition budget.

## 2. Problem Formulation

Consider a benchmark $X$ with $N$ prompts, $X = \{x_n\}_{n=1}^N$. When evaluating a model $m$ on this benchmark, we obtain

---

[1]Amazon Web Service. Correspondence to: Yang Li <ylizam@amazon.com>.

*Proceedings of the $42^{nd}$ International Conference on Machine Learning*, Vancouver, Canada. PMLR 267, 2025. Copyright 2025 by the author(s).

evaluation scores $Y_m = \{y_{mn}\}_{n=1}^N$. A leaderboard for this benchmark typically contains evaluation scores for $M$ models, denoted as $\{Y_m\}_{m=1}^M$. These scores may be of mixed types - for instance, some datasets might report binary accuracy while others use continuous metrics like F1 scores.

For a new model $m'$ to be evaluated, our goal is to acquire evaluation scores for only a subset of prompts $Y_{m'}^{(o)} = \{y_{m'o} : o \subseteq \{1, \ldots, N\}\}$ while predicting the scores $Y_{m'}^{(u)} = \{y_{m'u} : u = \{1, \ldots, N\} \setminus o\}$ for the remaining prompts. The key challenge is capturing the dependencies across prompts to ensure accurate prediction of the unobserved scores. We model these dependencies through the conditional distribution $p(Y_{m'}^{(u)} | Y_{m'}^{(o)}, X)$. Since the set of prompts to acquire their scores is not predefined, we must estimate $p(Y_{m'}^{(u)} | Y_{m'}^{(o)}, X)$ for all possible subsets $u$ and $o$.

Given a budget $K$ for acquiring evaluation scores and the above generative model, our objective is to find an optimal subset $o^* \subseteq \{1, \ldots, N\}$, where $|o^*| = K$, such that:

$$o^* = \arg\max_{o \in \mathcal{P}([N], K)} p(Y_{m'}^{(u)} | Y_{m'}^{(o)}, X), \qquad (1)$$

where $\mathcal{P}([N], K)$ represents all subsets of $\{1, \ldots, N\}$ with cardinality $K$. Note that this optimal subset could be different for each model being evaluated.

This objective presents two main challenges. First, the values $Y_{m'}^{(u)}$ for a test model $m'$ are unknown before acquisition, making direct optimization impossible. Second, the number of possible subsets grows combinatorially with $N$, making exhaustive search infeasible. Our solution addresses these challenges through an active learning inspired approach that iteratively selects prompts based on observed scores.

## 3. Method

### 3.1. Modeling Dependencies via Neural Processes

In this section, we aim to capture the dependencies across evaluation prompts by modeling the conditional distribution $p(Y_m^{(u)} | Y_m^{(o)}, X)$. We represent the relationship between the prompts and their evaluation scores as a stochastic process $\mathcal{F} : \mathcal{X} \to \mathcal{Y}$, where $\mathcal{X}$ and $\mathcal{Y}$ denote the spaces of prompts and their corresponding evaluation scores, respectively. Evaluating on benchmark $X$ can be interpreted as observing finite-dimensional marginal distributions of this stochastic process. Specifically, evaluation scores $Y_m$ represent function values $\{f_m(x_n)\}_{n=1}^N$ for a particular function, $f_m$, sampled from the distribution of functions.

Neural Processes (NPs) (Garnelo et al., 2018b;a; Kim et al., 2019) provide a flexible and scalable approach to modeling such stochastic processes. They combine the strengths of neural networks and Gaussian Processes to predict outputs for new inputs by conditioning on a set of context points.

Specifically, the function $f_m$ is implicitly parameterized by a latent vector $z_m$, and the generative model then follows

$$p(Y_m^{(u)} | Y_m^{(o)}, X) = \int p(z_m | Y_m^{(o)}, X) p(Y_m^{(u)} | z_m, Y_m^{(o)}, X) dz_m. \qquad (2)$$

Since the integration is over a high dimensional latent space, we optimize the evidence lower bound (ELBO) following variational autoencoder (VAE) (Kingma & Welling, 2013)

$$\log p(Y_m^{(u)} | Y_m^{(o)}, X)$$
$$\geq \mathbb{E}_{q(z_m | Y_m^{(u)}, Y_m^{(o)}, X)} \left[ \log \frac{p(Y_m^{(u)} | z_m, Y_m^{(o)}, X) p(z_m | Y_m^{(o)}, X)}{q(z_m | Y_m^{(u)}, Y_m^{(o)}, X)} \right], \qquad (3)$$

where $q(z_m | Y_m^{(u)}, Y_m^{(o)}, X)$ and $p(z_m | Y_m^{(o)}, X)$ represent the posterior and prior distributions, respectively.

To ensure our model represents a valid stochastic process, we adhere to the conditions stated by the Kolmogorov Extension Theorem (Oksendal, 2013): (finite) exchangeability and consistency. The exchangeability condition requires that the joint distribution $p(Y_m)$ remain unchanged under permutations of its elements, which we satisfy by using permutation invariant networks to parameterize both the prior and posterior distributions. The consistency demands that marginalizing out part of $Y_m$ yields the same distribution as that defined on the original prompt $x_n$. This condition is met when the approximate posterior equals the true posterior, which we achieve by training the model with sufficient data from diverse model evaluations so that the lower bound approaches the actual likelihood. In order to handle textual prompts, we utilize a pretrained embedding model to represent each prompt as a $\mathbb{R}^d$ vector. Please see Sec. B for implementation details.

### 3.2. Evaluation Acquisition Policy

Given the generative model across subsets of evaluation prompts, we now develop acquisition policies to select an optimal subset of prompts for acquiring their true evaluation scores, while the remaining scores will be predicted by the conditional $p(Y_m^{(u)} | Y_m^{(o)}, X)$.

#### 3.2.1. RANDOM POLICY

A random acquisition policy selects a subset of size $K$ at random to acquire the evaluation scores. Here, we consider two variants: **Uniform Sampling** and **Stratified Sampling** (Perlitz et al., 2023). Uniform Sampling selects $K$ prompts uniformly from $X$, while Stratified Sampling considers the size of different datasets and ensures each dataset is equally represented. The stratified sampling has been verified effective on HELM benchmark (Perlitz et al., 2023).

#### 3.2.2. STATIC POLICY

A static acquisition policy determines the set of prompts to be evaluated beforehand, and each model to be evaluated

acquires the evaluation scores on the same set of prompts. We assess the following two types of static policies:

**Clustering** Given the embedding for each prompt, we group them into $K$ cluster, then select one prompt in each cluster that is closest to the cluster centroid. We denote this approach as **Clustering-Embed**. Instead of using the pretrained sentence embedding, we can use the learned embedding from an Item Response Theory (IRT) model (Hambleton & Swaminathan, 2013; Embretson & Reise, 2013), which represents the difficulty and discriminability of each prompt. The **Clustering-IRT** method, proposed in (Polo et al., 2024), has been successfully applied on several public LLM benchmarks. Inspired by (Vivek et al., 2023), which selects representative examples by clustering based on prediction confidence, **Clustering-Score** groups prompts based on their evaluation scores on the training set. Each prompt $x_n$ is represented by a vector of evaluation scores, with the size of the vector corresponding to the number of evaluated models in the training set. The Clustering-Score method has been used as a baseline in (Polo et al., 2024).

**Combinatorial Optimization** Given the model $p(Y_m^{(u)} \mid Y_m^{(o)}, X)$, a static acquisition policy can be derived by searching over the training set to find the optimal subset of prompts that gives the most accurate prediction of the remaining prompts. This is a typical combinatorial optimization problem, which is NP-Hard. Here, we employ a sequential approach that selects one prompt at a time until $K$ prompts are selected. Starting from an empty set $o = \emptyset$, the next prompt $i \in u := \{1, \ldots, N\} \setminus o$ is chosen to minimize the prediction error over the training set, i.e.,

$$i = \arg\min_{i' \in u} \mathbb{E}_{Y_m \sim p_{\mathcal{D}}} \mathbb{E}_{\hat{Y}_m^{(u')}} \|\hat{Y}_m^{(u')} - Y_m^{(u')}\|^2, \quad (4)$$

where $o' = o \cup \{i'\}$ and $u' = u \setminus \{i'\}$ denote the updated observed and unobserved set after acquiring evaluation scores for prompt $i'$. $\hat{Y}_m^{(u')}$ indicates predicted scores sampled from the neural process model $p(Y_m^{(u')} \mid Y_m^{(o')}, X)$. The expectation is estimated by Monte Carlo sampling. For notation simplicity, the above equation computes the mean squared error on prompts $u'$; however, in practice, different datasets may use different metrics. Additionally, these differences may be weighted depending on the dataset size. Please refer to Algorithm 3 for pseudo-code of the selection process. Note that this approach has a complexity of $O(KMN)$, which could be prohibitive when the benchmark is large.

### 3.2.3. DYNAMIC POLICY

Instead of acquiring the same set of evaluation scores for each model, we propose dynamically acquiring adaptive subsets for different models, a method we term Active Evaluation Acquisition (AEA). This approach tailors the selection of prompts to each model's specific strengths and weaknesses, providing a more accurate and efficient evaluation.

---

**Algorithm 1** Active Evaluation Acquisition

**Require:** Budget $K$, model $m$ to be evaluated, Neural Process $p$
1: $o = \emptyset, Y_m^{(o)} = \emptyset, u = \{1, \ldots, N\}$
2: **while** $|o| < K$ **do**
3:  Select prompt $i$ according to (5), (6), or (9)
4:  Evaluate model $m$ on prompt $i$ to get evaluation score $Y_m^{(i)}$
5:  $o = o \cup \{i\}, Y_m^{(o)} = Y_m^{(o)} \cup \{Y_m^{(i)}\}, u = u \setminus \{i\}$
6: **end while**
7: Predict the evaluation scores for the remaining prompts $Y_m^{(u)} \sim p(Y_m^{(u)} \mid Y_m^{(o)}, X)$

---

Dynamic acquisition sequentially acquires evaluation scores and simultaneously refines the uncertainty of predictions, enabling real-time adaptation based on observed performance. AEA reduces redundancy by avoiding predictable evaluations and focusing resources on the most informative prompts. Please refer to Algorithm 1 for pseudo-code of the active acquisition process.

**Uncertainty Sampling** Inspired by uncertainty sampling method widely used in active learning literature (Ren et al., 2021; Yang et al., 2015; Raj & Bach, 2022), where the most uncertainty data point under the current predictor is chosen to query its label, we select the next prompt to be evaluated based on the uncertainty of $p(Y_m^{(i)} \mid Y_m^{(o)}, X)$. Here, $o$ contains the evaluated prompts so far, and $i \in u$ is one of the candidate prompts to be selected. We choose the prompt with the highest entropy:

$$i = \arg\max_{i \in u} H(Y_m^{(i)} \mid Y_m^{(o)}, X). \quad (5)$$

In practice, we estimate the entropy by sampling multiple times and computing the sample variance.

**Information Gain** Given the latent variable based neural process model (2), where the latent variable parameterizes the stochastic process, a straight-forward acquisition policy is to select the prompt that provides the most information about the latent variable $z_m$. We use the conditional mutual information to measure the amount of information:

$$
\begin{aligned}
i &= \arg\max_{i \in u} I(Y_m^{(i)}; z_m \mid Y_m^{(o)}, X) \\
&= \arg\max_{i \in u} \left[ H(z_m \mid Y_m^{(o)}, X) - \mathbb{E}_{\hat{Y}_m^{(i)}} H(z_m \mid \hat{Y}_m^{(i)}, Y_m^{(o)}, X) \right] \\
&= \arg\min_{i \in u} \mathbb{E}_{\hat{Y}_m^{(i)} \sim p(Y_m^{(i)} \mid Y_m^{(o)}, X)} H(z_m \mid \hat{Y}_m^{(i)}, Y_m^{(o)}, X).
\end{aligned}
$$
$$(6)$$

The third equation follows because the observed set $o$ is the same for any candidate $i \in u$. The expectation is estimated by Monte Carlo sampling. Note that the entropy is estimated based on predicted $Y_m^{(i)}$ rather than the true evaluation score as the true score is unknown before acquisition. At each acquisition step, the entropy must be estimated for each candidate prompt $i \in u$. Therefore, the total complexity is $O(KN)$, which could be prohibitive for large benchmarks.

**Reinforcement Learning** The active acquisition process can be formulated as a Markov decision process (MDP), where the state consists of the currently evaluated prompts and their scores, and the action space contains the remaining prompts to be evaluated. To solve the MDP, a reinforcement learning agent sequentially acquires new evaluation scores based on the current state. After acquiring evaluation score for prompt $i$, the current state transitions to a new state as follows: $o \xrightarrow{i} o \cup \{i\}, Y_m^{(o)} \xrightarrow{i} Y_m^{(o)} \cup \{Y_m^{(i)}\}$. When the agent acquires evaluation scores for $K$ prompts, the acquisition process terminates, and the agent receives a reward based on the prediction accuracy for the remaining prompts. In a basic setup, the agent would only receive a reward at the end of the acquisition process (after selecting K prompts) based on the prediction accuracy for the remaining prompts. However, this delayed reward structure poses a typical temporal credit assignment problem, which complicates the learning of an effective agent, especially when the trajectory is long (Minsky, 1961; Sutton, 1988). To address this issue, we design an intermediate reward function that provides immediate feedback after each acquisition action $i$. Specifically, the reward measures the improvement in prediction accuracy per unobserved prompt:

$$r(o, i) = \frac{\mathbb{E}_{\hat{Y}_m^{(u)}} \|\hat{Y}_m^{(u)} - Y_m^{(u)}\|^2}{|u|} - \frac{\mathbb{E}_{\hat{Y}_m^{(u')}} \|\hat{Y}_m^{(u')} - Y_m^{(u')}\|^2}{|u'|}, \quad (7)$$

where $o' = o \cup \{i\}$ and $u' = u \setminus \{i\}$ denote the observed and unobserved sets after acquiring prompt $i$. The expectations are taken over predicted evaluation scores sampled from the neural process model, with $\hat{Y}_m^{(u)} \sim p(Y_m^{(u)} \mid Y_m^{(o)}, X)$ before acquiring prompt $i$ and $\hat{Y}_m^{(u')} \sim p(Y_m^{(u')} \mid Y_m^{(o')}, X)$ after the acquisition. Note that this intermediate reward follows the potential function structure (Ng et al., 1999), therefore, it will not change the optimal policy.

Given this reward structure, the goal of the agent is to learn a policy $\pi^*$ that maximizes the expected cumulative reward over the trajectory of $K$ acquisitions:

$$\pi^* = \arg\max_\pi \mathbb{E}_{\tau \sim \pi} \left[ \sum_{t=1}^{K} r(o_t, i_i) \right], \quad (8)$$

where $\tau = (o_1, i_1, ..., o_K, i_K)$ represents a trajectory of states and actions under policy $\pi$, and $r(o_t, i_t)$ is the reward received after selecting prompt $i_t$ given the current observed set $o_t$. Note that the reward computation is only required during training when we have access to all the evaluation scores. During testing, the next prompt to be evaluated will be directly selected by the policy:

$$i = \arg\max_{i \in u} \pi^*(i \mid Y_m^{(o)}, X). \quad (9)$$

Since the policy network has constant computational cost, the total complexity of the acquisition process remains $O(K)$ regardless of the benchmark size.

In addition to providing intermediate rewards, we propose using the neural process to assist the agent with auxiliary information. Specifically, the neural process can predict the evaluation scores for unobserved prompts based on the observed scores in the current state. By sampling multiple times, the neural process can inform the agent about the uncertainties of these unobserved scores. The predicted scores and their uncertainties on the unobserved prompts allow the agent to anticipate future states and guide its exploration. For instance, if the neural process is very confident about the score of a currently unobserved prompt, then acquiring its real score would be redundant. The auxiliary information helps the agent make more informed decisions about which prompts to evaluate next, improving the efficiency and accuracy of the active acquisition process.

### 3.3. Cold Start Problem

As language models advance with emergent capabilities, benchmarks must expand with new prompts for which no evaluation scores are initially available. This cold start problem introduces two key challenges: generalizing the neural process to predict scores on new prompts, and determining which new prompts to select.

To address neural process generalization, we employ a semi-supervised approach where new prompts are treated as unlabeled data. Specifically, we incorporate predicted scores with high confidence into the training process through pseudo-labeling (Lee et al., 2013; Xie et al., 2020; Du et al., 2020). In our preliminary experiments, we also tested several common semi-supervised learning approaches, including entropy minimization (Grandvalet & Bengio, 2004) and consistency regularization (Tarvainen & Valpola, 2017), but found pseudo-labeling consistently performed best. A systematic exploration of semi-supervised neural process training remains as future work.

While static policies are limited by their reliance on historical scores, our dynamic approach enables continuous adaptation through sequential active acquisition. The key to handling previously unseen prompts lies in our novel policy architecture that explicitly incorporates prompt representations. At each acquisition step, the policy network $h$ takes two inputs: the acquired scores $Y_m^{(o)}$ and the embeddings of candidate prompts to be evaluated, where these embeddings are shared with the neural process model. The network outputs a vector in the same space as the prompt embeddings, and the selection probability is determined through inner products:

$$\pi(i \mid Y_m^{(o)}, X) = \frac{e^{a_i \cdot h(Y_m^{(o)}, X^{(o)}, \{a_i\}_{i \in u})}}{\sum_{i \in u} e^{a_i \cdot h(Y_m^{(o)}, X^{(o)}, \{a_i\}_{i \in u})}}, \quad (10)$$

where $\{a_i\}_{i \in u}$ denote the embeddings of candidate prompts. This dot-product architecture enables the policy to handle

arbitrary action spaces while maintaining awareness of available actions (Jain et al., 2020). Please refer to Appendix E for implementation details.

## 4. Related Works

**Active Learning** Active learning (Fu et al., 2013; Konyushkova et al., 2017; Yoo & Kweon, 2019) addresses the problem of having a learner select specific examples to query an oracle for their labels, with the goal of learning a better model using as few labeled examples as possible. In contrast, our proposed AEA framework focuses on evaluating a model with fewer examples to accurately predict the evaluation scores for the remaining examples.

**Active Testing** Active testing (Kossen et al., 2021) reduces the labeling cost by selectively choosing test points to label, ensuring sample-efficient model evaluation. This approach has been adapted LLM evaluation (Huang et al., 2024) and preference data selection (Ashury-Tahan et al., 2024). While this aligns with the overarching goal of efficient evaluation, our work specifically targets reducing the cost of running evaluations on a large number of prompts, rather than minimizing labeling costs.

**Computerized Adaptive Testing** IRT provides a principled framework for adaptive testing (Liu et al., 2024), selecting test items based on their difficulty and discrimination parameters. Zhuang et al. (2023) further applied this technique to LLM evaluation. While IRT methods offer interpretable parameters and theoretical guarantees, they typically assume item independence. Our approach differs by explicitly modeling dependencies between prompts through a neural process and learning adaptive selection strategies via RL, rather than relying on static item parameters.

**Efficient LLM Benchmarking** As LLMs continue to develop and scale, ongoing efforts aim to create benchmarks that comprehensively assess their capabilities. A notable trend in these benchmarks is their evolution from single-task assessments (Bowman et al., 2015; Rajpurkar et al., 2016) to multi-task benchmarks (Wang et al., 2018; 2019), and ultimately to massively multi-task evaluations (Srivastava et al., 2022; Liang et al., 2022; Hendrycks et al., 2020). The ever-increasing evaluation cost has encouraged researchers to develop efficient evaluation approaches. BIG-bench Lite (Srivastava et al., 2022) and BIG-bench Hard (Suzgun et al., 2022) evaluate on a subset of BIG-bench tasks, and Ye et al. (2023) propose clustering BIG-bench tasks and selecting the examples that are closest to cluster centers. Perlitz et al. (2023) found that the model rankings on HELM can be accurately obtained by evaluating only a fraction of the examples. Vivek et al. (2023) propose clustering the evaluation examples based on the uncertainty of model predictions, while Polo et al. (2024) suggest clustering examples based on learned features from an IRT model. In this work, we

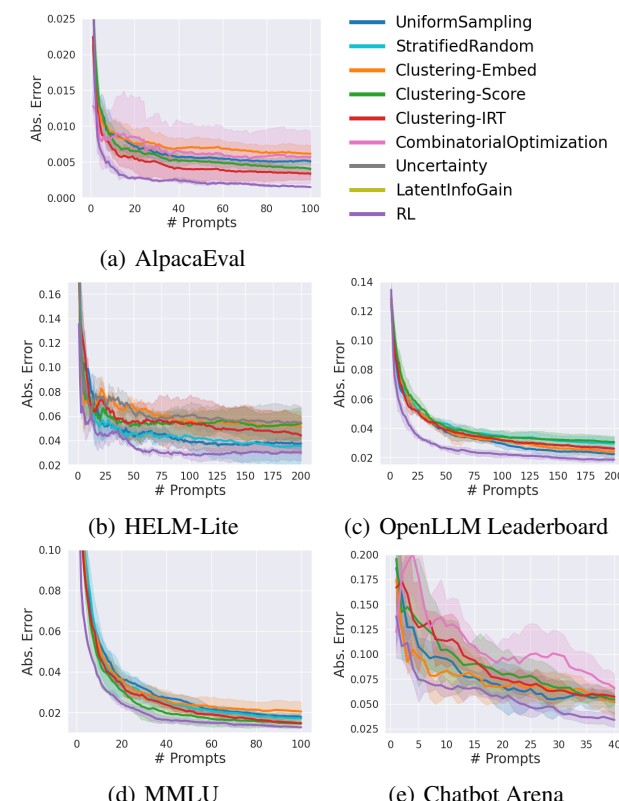

(a) AlpacaEval

(b) HELM-Lite

(c) OpenLLM Leaderboard

(d) MMLU

(e) Chatbot Arena

*Figure 1.* Experiment results on five LLM benchmarks, with shaded areas indicating the standard deviation over three runs.

comprehensively assess these methods and further propose actively selecting evaluation examples.

## 5. Experiments

In this section, we assess various evaluation acquisition policies on several popular LLM benchmarks. We divide the available leaderboard scores into training and test splits. The training split is used to fit the neural processes model, capturing the dependencies across prompts. The acquisition policies are executed for each model in the test split to acquire the evaluation scores for a subset of prompts. The evaluation scores on the remaining prompts are predicted based on the corresponding neural process model. The final score for each benchmark is computed as a weighted average across datasets, and we report the absolute differences between the predicted scores and the actual scores. Please see Appendix F for details.

We conduct experiments on five popular LLM benchmarks: HuggingFace Open LLM Leaderboard (Beeching et al., 2023), MMLU (Hendrycks et al., 2020), HELM-Lite (Liang et al., 2022), AlpacaEval 2.0 (Li et al., 2023), and Chatbot Arena (Zheng et al., 2024). For each benchmark, we divide its leaderboard into training and testing splits based on models rather than prompts. When evaluation dates are available

(e.g., for Open LLM Leaderboard), we use chronological splitting to ensure testing is performed on newer models, better simulating real-world scenarios where we evaluate newly released LLMs. For benchmarks without temporal information, we use random splitting. Detailed descriptions of these benchmarks can be found in Appendix F.

**Results** Figure 1 presents the main experimental results on 5 LLM benchmarks. We conduct experiments with 3 random seeds for each benchmark and plot the average performance and standard deviation throughout the acquisition process. Prompt embeddings are obtained using the SFR embedding model (Meng et al., 2024). For the static clustering based policies, since the selected prompts do not have an inherent order, the acquisition process shuffles the selected prompts at random. For the AlpacaEval and Chatbot Arena benchmarks, stratified random sampling is equivalent to uniform sampling since there are only one dataset in each benchmark. Combinatorial optimization is too expensive to run for HELM-Lite, HuggingFace Open LLM Leaderboard, and MMLU due to the large number of prompts. We found that uncertainty and information gain based policies consistently fail to explore the action space, leading to worse overall performance. To avoid cluttering the plots, results for uncertainty sampling and information gain based policies are moved to the appendix. Please refer to Appendix F for more analysis.

For all benchmarks, our proposed RL-based acquisition policy achieves the best performance with the lowest acquisition budget, demonstrating its superior ability to select informative prompts and accurately estimate benchmark performance. Notably, our method achieves dramatic reductions in required evaluations – requiring only 35 prompts to match the accuracy of random sampling with 100 prompts on MMLU, and similar efficiency gains across other benchmarks (75% reduction on HELM-Lite, 92% on AlpacaEval, see Sec. F.5 for detailed analysis). The stratified random sampling policy performs similarly to uniform sampling. Interestingly, the Clustering-Embed policy does not outperform the random selection, indicating that the similarity in prompt embedding does not always translate to the similarity in evaluation scores. Among the three clustering-based policies, none consistently outperforms the others. On AlpacaEval, HELM-Lite, and MMLU, the policies that utilize the evaluation scores (i.e., Claustering-Score and Clustering-IRT) perform better, while on the Open LLM Leaderboard and Chatbot Arena, Clustering-Embed perform better. The combinatorial optimization based policy does not perform well, even on the two small benchmarks where it is computationally feasible. We attribute this to a potential distribution shift between the models used for training and those used for testing, suggesting that the static policy optimized on training models does not generalize well to new models during testing. Please see Sec. F.3 and F.4 for additional

*Table 1.* Comparison of our RL-based acquisition policy with Tiny-Benchmarks (TB) (Polo et al., 2024), using selected prompts to predict evaluation scores with either the IRT model from TB or our NP model. The metric is the absolute prediction error.

|                    |    | IRT++              | NP                 |
|--------------------|----|--------------------|--------------------|
| AlpacaEval         | TB | $0.027 \pm 0.002$  | $0.003 \pm 0.001$  |
| (K=100)            | RL | **$0.014 \pm 0.005$** | **$0.001 \pm 0.000$** |
| MMLU               | TB | **$0.022 \pm 0.000$** | $0.016 \pm 0.000$  |
| (K=100)            | RL | $0.028 \pm 0.002$  | **$0.013 \pm 0.000$** |
| Open LLM           | TB | $0.023 \pm 0.002$  | $0.022 \pm 0.004$  |
| (K=200)            | RL | **$0.019 \pm 0.001$** | **$0.018 \pm 0.001$** |

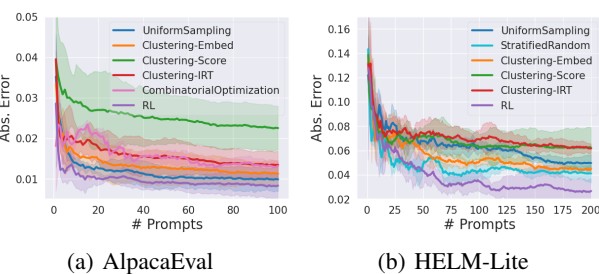

(a) AlpacaEval       (b) HELM-Lite

*Figure 2.* Evaluate the situation with model bias, where test models are from different model families compared to the training models.

results and analysis.

Additionally, we compare our methods with tinybenchmarks (Polo et al., 2024). Given the selected subsets from tinybenchmarks, we predict the final benchmark performance using both the IRT++ models provided by tinybenchmark [1] and our neural process models. Conversely, we also evaluate our prompt selections using both models. Table 1 compares the prompt selections from our proposed RL policy with those from tinybenchmark. The original tinybenchmark select 600 prompts for Huggingface Open LLM Leaderboard, but in our comparison, we select 200 prompts to ensure a fair comparison with our RL policy. The results show that for both prompt selections, using NP produces better benchmark performance estimates, indicating that our NP model better captures the dependencies and predicts the missing evaluation scores. Given a fixed prediction model (either IRT++ or NP), our RL-based acquisition policy achieves lower error compared to the prompt selections from tinybenchmark, demonstrating that our RL-based policy is more effective at selecting the informative prompts.

**Model Bias** An important aspect of efficient benchmarking strategies is robustness to model bias. To accurately evaluate future models, which may differ significantly from previously seen models, the strategy must accurately measure model capabilities based on the selected prompts. Our train-test splits based on date for MMLU and Open LLM

---

[1]https://github.com/felipemaiapolo/tinyBenchmarks/tree/main

*Table 2.* Comparison of the final benchmark performance estimation methods. **w/ pred** indicate the proposed method where the neural process is used to predict the missing evaluation scores. **w/o pred** indicates the baseline where final performance is an aggregation of the acquired evaluation scores.

| | | AlpacaEval (K=100) | HELM-Lite (K=200) | Open LLM (K=200) | MMLU (K=100) | Chatbot Arena (K=40) |
|---|---|---|---|---|---|---|
| Uniform | w/ pred | 0.005 ± 0.000 | 0.038 ± 0.005 | 0.022 ± 0.002 | 0.018 ± 0.001 | 0.052 ± 0.010 |
| | w/o pred | 0.012 ± 0.001 | 0.079 ± 0.008 | 0.043 ± 0.003 | 0.042 ± 0.003 | 0.036 ± 0.009 |
| S-Rand | w/ pred | - | 0.035 ± 0.012 | 0.030 ± 0.003 | 0.017 ± 0.002 | - |
| | w/o pred | - | 0.072 ± 0.012 | 0.023 ± 0.001 | 0.038 ± 0.001 | - |
| C-Embed | w/ pred | 0.006 ± 0.001 | 0.051 ± 0.010 | 0.024 ± 0.002 | 0.020 ± 0.004 | 0.052 ± 0.014 |
| | w/o pred | 0.023 ± 0.008 | 0.116 ± 0.003 | 0.029 ± 0.000 | 0.029 ± 0.000 | 0.032 ± 0.003 |
| C-Score | w/ pred | 0.004 ± 0.001 | 0.054 ± 0.010 | 0.031 ± 0.003 | 0.014 ± 0.002 | 0.054 ± 0.004 |
| | w/o pred | 0.141 ± 0.011 | 0.051 ± 0.017 | 0.086 ± 0.002 | 0.048 ± 0.002 | 0.037 ± 0.011 |
| C-IRT | w/ pred | 0.003 ± 0.001 | 0.044 ± 0.013 | 0.026 ± 0.001 | 0.015 ± 0.001 | 0.057 ± 0.002 |
| | w/o pred | 0.069 ± 0.003 | 0.060 ± 0.014 | 0.037 ± 0.003 | 0.041 ± 0.006 | 0.042 ± 0.003 |
| RL | w/ pred | 0.001 ± 0.000 | 0.030 ± 0.005 | 0.018 ± 0.001 | 0.013 ± 0.000 | 0.034 ± 0.006 |
| | w/o pred | 0.064 ± 0.006 | 0.081 ± 0.019 | 0.063 ± 0.018 | 0.050 ± 0.006 | 0.045 ± 0.007 |

Leaderboard potentially evaluate this situation since model performance tends to improve over time. To further evaluate the performance in the presence of model bias, we divide the models on the AlpacaEval and HELM-Lite leaderboards based on their organizations. For HELM-Lite, we use proprietary models, such as GPT-4 (Achiam et al., 2023) and Claude (Anthropic, 2024), for training and test on open-source models, such as LLaMA (Touvron et al., 2023) and Mistral (Jiang et al., 2023). For AlpacaEval, we do the opposite, using open-source models for training and proprietary models for testing.

Figure 2 presents evaluation results on these two benchmarks with model bias. Firstly, static policies, especially Clustering-Score and Clustering-IRT that depend on evaluation scores from the training models, do not perform well. Secondly, although random policies do not suffer from model bias, they cannot leverage dependencies across prompts, leading to lower overall performance. In contrast, our RL-based dynamic acquisition policy can effectively exploit the dependencies across prompts even for models that is significantly different from the models it has seen before. However, we notice that the existence of model bias makes the problem harder to solve. Compared to Fig. 1 on the same benchmark, even for our RL-based policy, it takes more acquisitions to achieve the same level of errors as in situations where no model bias exists. In practice, a continual learning framework, where the NP model and the acquisition policies are jointly adapted to the newly added models, might be necessary. We leave this for future works.

**Cold Start Problem** To evaluate the cold start scenario, we create a synthetic benchmark using MMLU by designating 15 subsets as cold start prompts. During the training of the neural process model and the acquisition policies, the evaluation scores on these 15 subsets are not available.

Although the evaluation scores are missing, we assume the prompts themselves are given, allowing random policies and Clustering-Embed static policy to be evaluated without any modifications. However, the Clustering-Score and Clustering-IRT policies will never acquire evaluation scores for these 15 subsets since these policies require access to the evaluation scores to determine whether a prompt will be acquired or not. On the other hand, dynamic acquisition policies can easily adapt to the cold start setting, as they acquire evaluation scores sequentially and actively.

Figure 3 presents the results on the synthetic cold start MMLU benchmark. The performance is evaluated over all 57 subsets during testing. As expected, the Clustering-Score and Clustering-IRT policies do not perform well in the cold start setting because the evaluation scores on the 15 left-out subsets are never acquired. The Clustering-Embed policy performs better than the other clustering based policies as it can select the cold start prompts by clustering based on their embeddings. The RL-based acquisition policy again achieves the best performance estimation. However, it is worth noting that the final estimated benchmark performance is not as accurate as in the fully observed setting (Fig. 1), indicating potential areas for future improvement to narrow the gap.

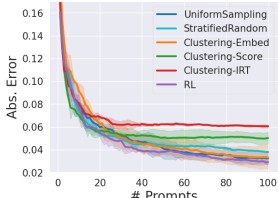

*Figure 3.* Evaluate the cold start problem on MMLU benchmark, where 15 subsets are left out as cold start prompts.

### 5.1. Ablation Studies

**Prediction Model** In the main experimental results, we run the acquisition policy to select a subset of prompts

*Table 3.* Comparison of different prompt embeddings.

| | | Uniform | C-Embed | C-Score | C-IRT | RL |
|---|---|---|---|---|---|---|
| AlpacaEval | SFR (4096) | $0.005 \pm 0.000$ | $0.006 \pm 0.001$ | $0.004 \pm 0.001$ | $0.003 \pm 0.001$ | $0.001 \pm 0.000$ |
| | E5 (4096) | $0.005 \pm 0.000$ | $0.006 \pm 0.001$ | $0.007 \pm 0.005$ | $0.004 \pm 0.001$ | $0.001 \pm 0.000$ |
| | BGE-large (1024) | $0.006 \pm 0.001$ | $0.006 \pm 0.002$ | $0.009 \pm 0.002$ | $0.007 \pm 0.001$ | $0.002 \pm 0.000$ |
| | BGE-small (384) | $0.009 \pm 0.002$ | $0.009 \pm 0.002$ | $0.038 \pm 0.031$ | $0.015 \pm 0.010$ | $0.005 \pm 0.002$ |
| MMLU | SFR (4096) | $0.018 \pm 0.001$ | $0.020 \pm 0.004$ | $0.014 \pm 0.002$ | $0.015 \pm 0.001$ | $0.013 \pm 0.000$ |
| | E5 (4096) | $0.018 \pm 0.002$ | $0.018 \pm 0.002$ | $0.014 \pm 0.003$ | $0.016 \pm 0.002$ | $0.014 \pm 0.003$ |
| | BGE-large (1024) | $0.029 \pm 0.003$ | $0.028 \pm 0.005$ | $0.027 \pm 0.005$ | $0.023 \pm 0.006$ | $0.023 \pm 0.003$ |
| | BGE-small (384) | $0.028 \pm 0.003$ | $0.023 \pm 0.006$ | $0.022 \pm 0.005$ | $0.022 \pm 0.005$ | $0.022 \pm 0.002$ |

*Table 4.* Contributions of auxiliary information and intermediate reward for our RL-based policy.

| | AlpacaEval | MMLU | Open LLM |
|---|---|---|---|
| PPO | $0.004 \pm 0.002$ | $0.017 \pm 0.004$ | $0.033 \pm 0.010$ |
| +auxiliary_info | $0.003 \pm 0.001$ | $0.016 \pm 0.003$ | $0.029 \pm 0.005$ |
| +interm_reward | $\mathbf{0.001 \pm 0.000}$ | $\mathbf{0.013 \pm 0.000}$ | $\mathbf{0.018 \pm 0.001}$ |

for acquiring their actual evaluation scores and then use a neural process model to predict the scores for the remaining prompts. However, an alternative method to estimate benchmark performance is to directly aggregate the acquired evaluation scores without relying on another model for prediction. The aggregation computes performance per dataset first and then averages across datasets. Table 2 compares these two estimation methods. The results show that the prediction model generally provides better benchmark performance estimation.

**Prompt Embedding**   Our approach utilizes a sentence embedding model to extract representations for the prompts. These representations are used both to train the neural process model and to build the acquisition policies. For the main results, we use the SFR embedding model (Salesforce/SFR-Embedding-Mistral) (Meng et al., 2024) to extract prompt representations. In Table 3, we present results using several other embedding models: E5, BGE-large, and BGE-small, corresponding to intfloat/e5-mistral-7b-instruct (Wang et al., 2023), BAAI/bge-large-en-v1.5 (Xiao et al., 2023), and BAAI/bge-small-en-v1.5 (Xiao et al., 2023), respectively. The results indicate that performance generally improves with more powerful embedding models that better distinguish text inputs[2]. Thus, exploring powerful embedding models is an important future direction.

**Auxiliary Information**   Our RL-based acquisition policy builds on PPO (Schulman et al., 2017) and leverages the neural process model to provide auxiliary information and

intermediate rewards. Table 4 illustrates the contributions of these components. The results clearly show that each component – both auxiliary information and the intermediate rewards – significantly enhances the acquisition policy, leading to better selection of informative prompts and more accurate benchmark performance estimation.

## 6. Conclusion

In this work, we present a novel approach for efficient LLM evaluation by leveraging dependency modeling and subset selection. Our key contributions include developing a generative model that captures dependencies across evaluation prompts and handles mixed-type evaluation scores, as well as proposing new subset selection policies based on these dependencies. Extensive experiments on multiple LLM evaluation benchmarks demonstrate the superiority of our RL-based acquisition policy in providing accurate benchmark performance estimation with minimal acquisition budget. Our results show that we can achieve the same level of accuracy while requiring only 35-75% of the evaluation prompts compared to random sampling across different benchmarks.

Our approach effectively addresses model bias and cold start scenarios, though performance in these challenging settings indicates room for improvement. Future work could explore integrating continual learning frameworks to enhance adaptation to new models and prompts. Additionally, leveraging more sophisticated embedding models and improving uncertainty estimation in the neural process could further boost performance. Our framework provides a foundation for making comprehensive LLM evaluation more accessible and efficient.

---

[2]At the time of writing this paper, the average scores from the MTEB English leaderboard. (https://huggingface.co/spaces/mteb/leaderboard) for these four models are: SFR (67.56), E5 (66.63), BGE-large (64.23), and BGE-small (62.17).

## Impact Statement

Our work aims to improve the efficiency of LLM evaluation, which has several important implications. By reducing the computational resources required for comprehensive model evaluation, our method can help decrease the environmental impact of AI development through reduced energy consumption. This is particularly relevant given the growing concerns about the carbon footprint of large-scale AI systems.

Additionally, more efficient evaluation methods could democratize LLM development by lowering the barrier to entry for researchers and organizations with limited computational resources. This could lead to increased diversity in AI research and development, as more groups would be able to rigorously evaluate their models without requiring extensive computational infrastructure or substantial API costs.

However, we acknowledge that easier model evaluation could potentially accelerate the development and deployment of LLMs, which carry their own ethical considerations and societal impacts. We encourage users of our method to carefully consider the broader implications of their work and to maintain high standards for model evaluation, even when using efficient sampling approaches.

Furthermore, while our method significantly reduces the number of required evaluations, it is important to note that this efficiency should not come at the cost of thorough model assessment, particularly for safety-critical applications or when evaluating potential harmful behaviors. We recommend using our method as part of a comprehensive evaluation strategy rather than as a complete replacement for thorough testing.

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

# A. Training and Deployment Procedure

## A.1. Overview

Our approach consists of two key components trained on historical evaluation data:

1. A neural process model that captures dependencies across prompts and predicts unobserved evaluation scores

2. A subset selection policy that determines which prompts to evaluate for new models

## A.2. Training Procedure

Given a benchmark leaderboard with $M$ models and $N$ prompts, we first divide the models into train and test splits. For each model $m$ in the training split, we have access to its complete evaluation scores $Y_m = \{y_{mn}\}_{n=1}^N$.

### A.2.1. NEURAL PROCESS TRAINING

During training:

- For each training iteration:
  - Sample a model $m$ from training split
  - Randomly partition $Y_m$ into observed set $Y_m^{(o)}$ and unobserved set $Y_m^{(u)}$
  - Optimize ELBO objective: maximize $\log p(Y_m^{(u)} \mid Y_m^{(o)}, X)$

- The trained model can predict scores for arbitrary subsets of prompts on unseen models

### A.2.2. SUBSET SELECTION POLICY TRAINING

We implement and compare several policies:

- Parameter-free policies:
  - **Random**: Selects prompts uniformly at random
  - **Clustering-Embed**: Groups prompts using pretrained embeddings
  - **Clustering-Score**: Uses training set scores as features for clustering
  - **Uncertainty Sampling**: Selects prompts with highest prediction uncertainty based on the neural process model
  - **Information Gain**: Maximizes expected information about latent variables using the above neural process model
  - **Combinatorial Optimization**: Greedily optimizes prediction accuracy based on the above neural process model

- Policies requiring training:
  - **Clustering-IRT**: Trains IRT model to learn prompt embeddings
  - **RL-based**: Learns policy through interaction with neural process model

## A.3. Deployment Procedure

For evaluating a new model:

1. Policy selects $K$ prompts for evaluation

2. Obtain actual evaluation scores for selected prompts

3. Neural process predicts scores for remaining prompts

4. Combine actual and predicted scores for final performance estimate

## B. Neural Process

For a benchmark $X$ with $N$ prompts, $X = \{x_n\}_{n=1}^N$, we first use a pretrained embedding model to extract the representations for each prompt. During training, given a model $m$ with evaluation scores $Y_m$, we randomly select a subset of scores $Y_m^{(o)}$ as observed and maximize the log-likelihood for the remaining scores $Y_m^{(u)}$ based on the equation (3). When $Y_m^{(u)}$ is too large to fit into memory, we further sample a smaller subset from $Y_m^{(u)}$. Due to the inherent permutation invariance of the neural process model, random sampling will not affect the learning of dependencies across prompts. Please see Algorithm 2 for training details of the neural process model.

---

**Algorithm 2** Neural Process Training

---

**Require:** Training set $\{Y_m\}_{m=1}^M$, Number of iterations $T$
 1: **for** $t = 1$ to $T$ **do**
 2:     Sample model $m$ from training set
 3:     Randomly partition $Y_m$ into $Y_m^{(o)}$ and $Y_m^{(u)}$
 4:     Optimize ELBO: $\mathcal{L} = \mathbb{E}_{q(z_m|Y_m^{(u)},Y_m^{(o)},X)}\left[\log \frac{p(Y_m^{(u)}|z_m,Y_m^{(o)},X)p(z_m|Y_m^{(o)},X)}{q(z_m|Y_m^{(u)},Y_m^{(o)},X)}\right]$
 5: **end for**

---

### B.1. Architecture

The neural process model consists of a prior network $p(z_m \mid Y_m^{(o)}, X)$, a posterior network $q(z_m \mid Y_m^{(u)}, Y_m^{(o)}, X)$, and a decoder $p(Y_m^{(u)} \mid z_m, Y_m^{(o)}, X)$. We generally follow the architecture of Attentive Neural Process (Kim et al., 2019), but replace the self-attention layer with a more memory-efficient Set Transformer layer (Lee et al., 2018). We also share the same network for both the prior and posterior. Before feeding the prompt embeddings into the prior/posterior network, we use an additional linear layer to reduce the dimensionality of the extracted representations. Similarly, the evaluation scores are passed through a linear layer to increase their dimensionality. We then concatenate the prompt representation with the score representation along the feature dimension and pass the concatenated set of vectors through a series of permutation equivariant Set Transformer layers. The outputs are then aggregated across the set elements to obtain a feature representation for the entire set. Following Set Transformer approach, we use learned pooling by multihead attention. The set representation is then passed through a linear linear to obtain the parameters for the latent distribution, which we assume to be Gaussian here. Please see Fig. 1(a) for an illustration of the prior/posterior network.

The decoder network $p(Y_m^{(u)} \mid z_m, Y_m^{(o)}, X)$ uses a cross-attention layer to produce a permutation equivariant representation for each prompt. In this layer, the query is the representation for $X$, the key is the representation for $X^{(o)}$, and the value is the permutation equivariant representation corresponding to $Y_m^{(o)}$ from the prior network. The permutation equivariant representation for each prompt is then concatenated with the prompt representation and the latent vector. These concatenated inputs are processed through a series of Set Transformer layers. The final outputs are then passed through a linear layer to predict the evaluation scores. Please see Fig. 1(b) for an illustration of the decoder network.

### B.2. Implementation

In order to handle textual prompts, we utilize a pretrained embedding model to represent each prompt as a $\mathbb{R}^d$ vector. During training, since the entire set $Y_m$ might be too large to fit into memory, we randomly sample two non-overlapping subsets from each model as $Y_m^{(o)}$ and $Y_m^{(u)}$, respectively. The prior and posterior distributions share the same network, but take different inputs. The prior takes in a set of x-y pairs from $Y_m^{(o)}$, i.e., $\{(x_o, y_{mo}) : o \subseteq \{1, \ldots, N\}\}$, while the posterior takes in a set of x-y pairs from both $Y_m^{(o)}$ and $Y_m^{(u)}$. Following the Attentive Neural Process (Kim et al., 2019), we implement the prior/posterior network using self-attention blocks to better capture the dependencies across set elements. To reduce memory usage, we use Set Transformer architecture (Lee et al., 2018), where each set element attends to a small set of learnable induced points instead of attending to all other elements directly. The decoder network $p(Y_m^{(u)} \mid z_m, Y_m^{(o)}, X)$ employs cross-attention, allowing each unobserved prompt to attend to the relevant observed prompts. According to De Finetti's Theorem (De Finetti, 1929), the likelihood over set elements $Y_m^{(u)}$ can be conditionally independent conditioned on the latent variable $z_m$. However, we still use a Set Transformer (Lee et al., 2018) to better capture the dependencies. Please refer to Appendix B for details of the model architecture.

## B.3. Mixed-type Evaluation Scores

The above architecture uses a linear layer to obtain the representation for the evaluation scores. However, the linear layer is not suitable for discrete scores. Instead, we use an Embedding layer to represent the categorical evaluation scores. When a benchmark contains mixed-type scores, meaning some datasets report real-valued metrics while others report discrete scores, we additionally include an embedding vector to indicate the metric types.

## B.4. Hyperparameters

Table B.1 summarizes the hyperparameters used for the neural process model for each dataset. For the HELM-Lite and Chatbot Arena benchmarks, due to their relatively small number of models with evaluation scores, a neural process model with set transformer layers can easily overfit the data. Therefore, we use linear layers instead of the set transformer layers. Note that we did not conduct a thorough hyperparameter search. It is possible to further improve the results with optimized hyperparameters.

*Table B.1.* Hyperparameters for the nueral process model.

|  | AlpacaEval | MMLU | Open LLM | HELM-Lite | Chatbot Arena |
|---|---|---|---|---|---|
| representation dimension for $x$ | 16 | 16 | 16 | 16 | 16 |
| representation dimension for $y$ | 16 | 16 | 16 | 16 | 16 |
| feature dimension for permutation equivariant layer | 32 | 32 | 32 | 32 | 16 |
| number of permutation equivariant layers for encoder | 1 | 2 | 2 | 1 | 1 |
| number of permutation equivariant layers for decoder | 1 | 2 | 2 | 1 | 1 |
| number of attention heads | 8 | 8 | 8 | N/A | N/A |
| number of induced points | 16 | 16 | 8 | N/A | N/A |
| latent dimension | 16 | 32 | 32 | 16 | 8 |

# C. Combinatorial Optimization based Acquisition Policy

Given the model $p(Y_m^{(u)} \mid Y_m^{(o)}, X)$, a static acquisition policy can be derived by searching over the training set to find the optimal subset of prompts that gives the most accurate prediction of the remaining prompts. This is a typical combinatorial optimization problem, which is NP-Hard. Here, we employ a sequential approach that selects one prompt at a time until $K$ prompts are selected. Starting from an empty set $o = \emptyset$, the next prompt $i \in u := \{1, \ldots, N\} \setminus o$ is chosen to minimize the prediction error over the training set, i.e.,

$$i = \arg\min_{i' \in u} \mathbb{E}_{Y_m \sim p_{\mathcal{D}}} \mathbb{E}_{\hat{Y}_m^{(u')} \sim p(Y_m^{(u')} \mid Y_m^{(o')}, X)} \|\hat{Y}_m^{(u')} - Y_m^{(u')}\|^2, \tag{C.1}$$

where $o' = o \cup \{i'\}$ and $u' = u \setminus \{i'\}$. We estimate the expectation by Monte Carlo sampling. For notation simplicity, the above equation computes the mean squared error on prompts $u'$; however, in practice, different datasets may use different metrics. Additionally, these differences may be weighted depending on the dataset size. Please refer to Algorithm 3 for pseudo-code of the selection process. Note that this approach has a complexity of $O(KMN)$, which could be prohibitive when the benchmark is large.

# D. Reinforcement Learning based Acquisition Policy

The acquisition policy determines the next prompt to acquire its evaluation score based on the current state, which includes the prompts $X^{(o)}$ and their scores $Y_m^{(o)}$ that have already been acquired. We further incorporate the candidate prompts $X^{(u)}$ into the policy inputs, i.e., $P(i \mid Y_m^{(o)}, X)$, so the policy has access to the action space. Including the candidate prompts in the inputs is crucial in the cold start setting since the action space differs between training and testing. Similar to the neural process model, the policy network employs two linear layers to obtain representations for both the prompts and the evaluation scores, which are then concatenated along the feature dimension. For the candidate prompts without available evaluation scores, we use a special embedding vector. Then, a permutation-invariant network processes the set of concatenated representations and outputs a aggregated representation for the entire set. We utilize the Set Transformer architecture for the permutation invariant network. Two branches of linear layers are added on top of the set representation for actor and critic, respectively. The actor branch outputs a vector with the same dimensionality as the prompt representations.

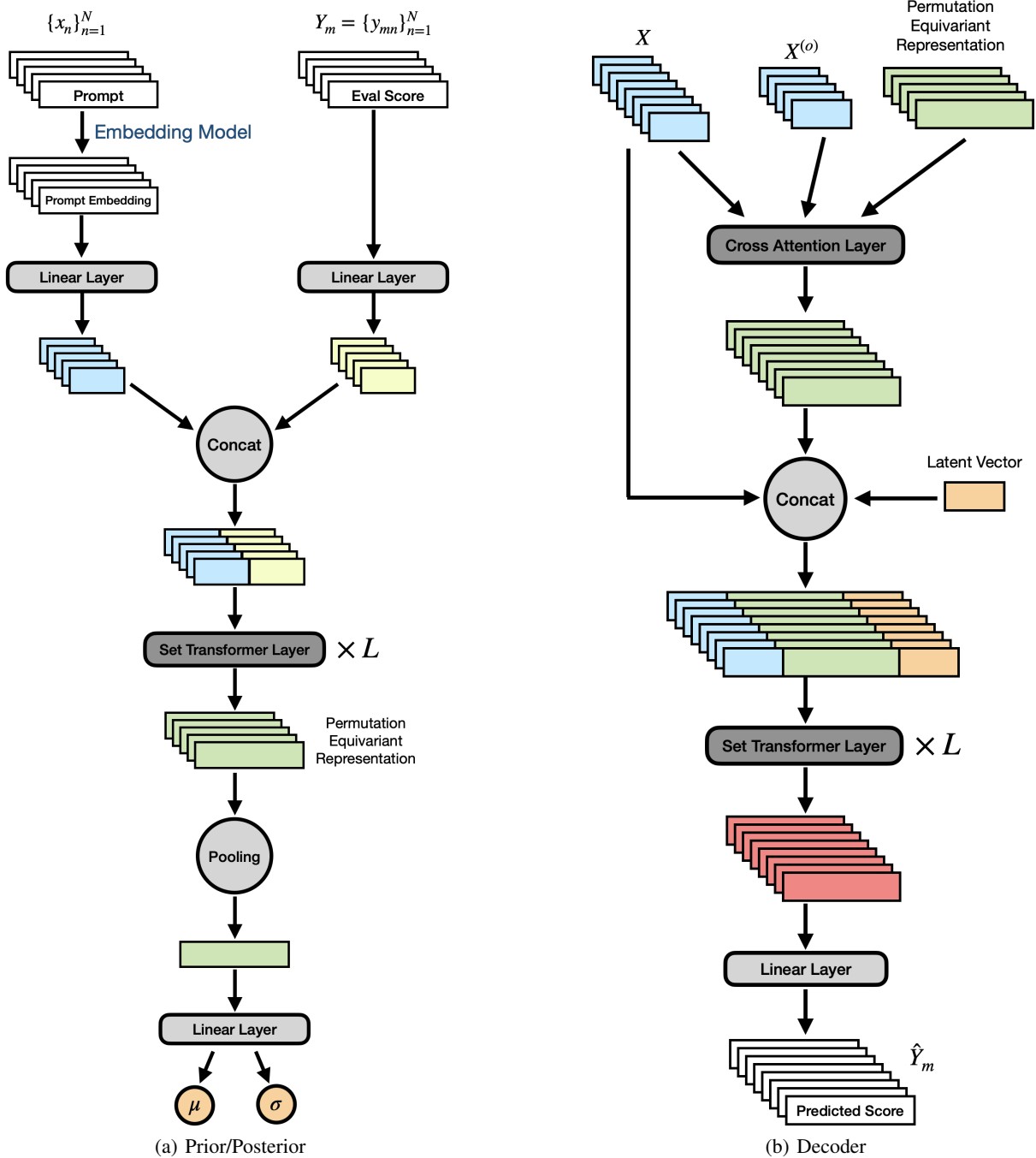

*Figure B.1.* The architecture of the neural process model.

---

**Algorithm 3** Static Evaluation Acquisition via Combinatorial Optimization

---

**Require:** Acquisition budget $K$, Training set $\mathcal{D}_{train}$, Number of samples $S$, Neural Process $p$

1:   $o = \emptyset, u = \{1, \ldots, N\}$
2: **while** $|o| < K$ **do**
3:     $L = \{\}$
4:    **for** $i' \in u$ **do**
5:      $o' = o \cup \{i'\}, u' = u \setminus \{i'\}$
6:      Sample $S$ predictions $\{\hat{Y}_{m,s}^{(u')}\}_{s=1}^{S}$ from $p(Y_m^{(u')} \mid Y_m^{(o')}, X)$ for each model $m$
7:      $L[i'] = \frac{1}{|\mathcal{D}_{train}| \times S} \sum_{m=1}^{|\mathcal{D}_{train}|} \sum_{s=1}^{S} \|\hat{Y}_{m,s}^{(u')} - Y_m^{(u')}\|^2$
8:    **end for**
9:     $i = \arg\min_{i' \in u} L[i']$
10:    $o = o \cup \{i\}, u = u \setminus \{i\}$
11: **end while**

---

The probability of selecting a prompt is proportional to the inner product of the output vector and the prompt representations. To prevent the policy from selecting duplicate prompts, the probability of the already selected prompts is manually set to zero. The critic branch outputs a scalar indicating the value estimation for the current state. Table D.1 summarizes the hyperparameters used for the policy network and PPO training process. We did not conduct hyperparameter optimization and used the same set of hyperparameters for all datasets. Further improvements are likely possible with hyperparameter optimization tailored to each dataset.

*Table D.1.* Hyperparameters for RL-based acquisition policy.

| | | |
|---|---|---|
| | representation dimension for $x$ | 16 |
| | representation dimension for $y$ | 16 |
| | feature dimension for permutation equivariant layer | 32 |
| Policy Network | number of permutation equivariant layers | 1 |
| | number of linear layers for actor | 1 |
| | number of linear layers for critic | 1 |
| | advantage $\lambda$ | 0.95 |
| | discount factor $\gamma$ | 0.99 |
| PPO | PPO clip range | [0.8, 1.2] |
| | entropy coefficient | 0.0 |

# E. Cold Start Problem

In the cold start setting, the benchmark is expanded with new prompts for which no evaluation scores are initially available for any model. That is, the original benchmark $X = \{x_n\}_{n=1}^{N}$ have evaluation scores $Y_m = \{y_{mn}\}_{n=1}^{N}$ for $M$ models, while a set of new prompts $X' = \{x_n\}_{n=N+1}^{N'}$ do not have any evaluation scores.

To enable the neural process model generalize to the newly added prompts, we propose a semi-supervised training procedure, where the new prompts are treated as unlabeled data. During training, we optimize the log-likelihood (3) for $X$ and $Y_m$. Simultaneously, we predict the evaluation scores for the new prompts $X'$ based on the current trained model. When the prediction is sufficiently accurate, meaning the uncertainty is lower than a predefined threshold, we add the predicted scores as synthetic training data to optimize the ELBO (3).

The RL policy in the cold start setting follows a similar architecture to Sec. D. To help the policy generalize to unseen prompts, we use the learned prompt representations from the neural process model and keep them fixed throughout the training process. Additionally, we found that entropy regularization over the actor distribution aids generalization, which is set to 0.001 in our experiments.

# F. Experiments

## F.1. LLM Leaderboard

We conduct experiments on 5 popular LLM benchmarks:

- **HuggingFace Open LLM Leaderboard** (Beeching et al., 2023) consists of 6 datasets with a total of 28,659 prompts. Evaluation scores include both binary accuracy and real-valued probabilities. We collect evaluation scores for 2,084 models and select 1,000 models for training based on their evaluation date. The most recently evaluated models are used for testing, simulating the real-world scenario.

- **MMLU** (Hendrycks et al., 2020) contains 57 datasets with a total of 14,042 multiple choice QA problems on different subjects. Evaluation scores are all binary accuracy. We collect evaluation scores for the same models from the Open LLM Leaderboard.

- **HELM-Lite** (Liang et al., 2022) include 10 datasets (each possibly containing several sub-datasets) with a total of 13,021 prompts. Evaluation scores include both binary exact match scores and real-values metrics such as F1 and BLEU. We collect evaluation scores for 33 models and randomly select 23 models for training since the evaluation does not have dates.

- **AlpacaEval 2.0** (Li et al., 2023) contains 805 prompts. For each model, the generations are compared to those of GPT-4 to compute the win rate. Although this benchmark is relatively small, it requires an expensive GPT-4 based judge, so reducing the number of API calls can significantly reduce the total evaluation cost. We collect evaluation scores for 130 models and randomly select 70% for training.

- **Chatbot Arena** (Zheng et al., 2024) is a popular human-annotated benchmark, where annotators interact with two anonymous models using the same prompts and declare a winner. We use the pairwise comparisons evaluated on the 80 MTBench prompts (Zheng et al., 2024). Although this benchmark is relatively small, human evaluation is expensive, so further reducing the evaluation prompts could lower costs. The annotations include comparisons over multiple turns, but we only use the annotations for the first turn here. Unlike other benchmarks where each model directly receives an evaluation score, this benchmark evaluates each pair from a set of 6 models. To create the train-test splits, we randomly select one of the six models, and all pairs that involve the selected model are included in the test split. Note that not all 80 prompts are annotated for each model pair. While our neural process model can handle missing data, the acquisition process must acquire the true score for any prompt the policy selects. To address the missing data during acquisition process, we use the trained neural process to predict the missing evaluation scores. We report win rate for this benchmark.

## F.2. Evaluation Procedure

For a model $m'$ to be evaluated, the acquisition policy determines a subset of prompts $X^{(o)}$ to acquire the true evaluation scores. The neural process model $p(Y_{m'}^{(u)} \mid Y_{m'}^{(o)}, X)$ predicts the evaluation scores for the remaining prompts. The benchmark performance is then estimated based on these predicted scores. For benchmarks with only one dataset, the benchmark performance is the average over all examples. For benchmarks with multiple datasets, the benchmark performance is averaged over the performance of each dataset. For example, the HuggingFace Open LLM Leaderboard consists of 6 datasets, so the benchmark performance is the average of the performance on these 6 datasets. The MMLU dataset further contains 57 subsets, so its performance is the average over these 57 subsets. For Chatbot Arena, we report win rate as the benchmark performance. For final evaluation results, we compute the absolute difference between the predicted benchmark performance and the real benchmark performance for each model in the test split and report the average absolute error over all models in the test split.

## F.3. Additional Results

Table F.1 presents the benchmark performance estimation errors for various acquisition policies across different LLM benchmarks. We conduct experiments with 3 random seeds for each benchmark and report the average estimation error and standard deviation under the specified acquisition budget. Prompt embeddings are obtained using the SFR embedding model. For the AlpacaEval and Chatbot Arena benchmarks, stratified random sampling is equivalent to uniform sampling since they only contain one dataset. Combinatorial optimization and information gain based policies are too expensive to run for HELM-Lite, HuggingFace Open LLM Leaderboard, and MMLU due to the large number of prompts in each benchmark.

*Table F.1.* Benchmark performance estimation error on each LLM benchmark. Lower is better.

|  | AlpacaEval (K=100) | HELM-Lite (K=200) | Open LLM (K=200) | MMLU (K=100) | Chatbot Arena (K=40) |
|---|---|---|---|---|---|
| Uniform | $0.005 \pm 0.000$ | $0.038 \pm 0.005$ | $0.022 \pm 0.002$ | $0.018 \pm 0.001$ | $0.052 \pm 0.010$ |
| S-Rand | - | $0.035 \pm 0.012$ | $0.030 \pm 0.003$ | $0.017 \pm 0.002$ | - |
| C-Embed | $0.006 \pm 0.001$ | $0.051 \pm 0.010$ | $0.024 \pm 0.002$ | $0.020 \pm 0.004$ | $0.052 \pm 0.014$ |
| C-Score | $0.004 \pm 0.001$ | $0.054 \pm 0.010$ | $0.031 \pm 0.003$ | $0.014 \pm 0.002$ | $0.054 \pm 0.004$ |
| C-IRT | $0.003 \pm 0.001$ | $0.044 \pm 0.013$ | $0.026 \pm 0.001$ | $0.015 \pm 0.001$ | $0.057 \pm 0.002$ |
| Comb-Optim | $0.006 \pm 0.003$ | - | - | - | $0.065 \pm 0.012$ |
| Uncertainty | $0.011 \pm 0.001$ | $0.055 \pm 0.013$ | $0.063 \pm 0.015$ | $0.050 \pm 0.003$ | $0.035 \pm 0.003$ |
| LatentInfoGain | $0.010 \pm 0.003$ | - | - | - | $0.066 \pm 0.025$ |
| RL | $\mathbf{0.001 \pm 0.000}$ | $\mathbf{0.030 \pm 0.005}$ | $\mathbf{0.018 \pm 0.001}$ | $\mathbf{0.013 \pm 0.000}$ | $\mathbf{0.034 \pm 0.006}$ |

The RL-based acquisition policy consistently achieves the lowest error across all benchmarks, indicating its superior ability to select informative prompts and accurately estimate benchmark performance. The stratified random sampling performs similarly to the uniform sampling, and these random acquisition policies generally are competitive, particularly because they are efficient and do not rely on any other models to determine the prompt selection.

The Cluster-Embed policy does not perform any better than the random selection, suggesting that the similarity in prompt embedding does not always correlate with the similarity in the evaluation scores. Utilizing evaluation scores for clustering shows mixed results. The Clustering-Score policy outperforms Clustering-Embed on AlpacaEval and MMLU but underperforms on HELM-Lite, Open LLM and Chatbot Arena benchmarks. Clustering based on IRT features generally provides better performance estimation since these features are learned to reflect the evaluation scores.

The combinatorial optimization based policy does not perform well, even on the two small benchmarks where it is computationally feasible. We attribute this to a potential distribution shift between the models used for training and those used for testing, suggesting that the static policy optimized on training models does not generalize well to new models during testing.

The uncertainty sampling based acquisition policy does not perform well across all benchmarks. Theoretically, the uncertainty sampling method requires a good estimation of the aleatoric uncertainty to perform well. However, in practice, the uncertainty from the neural process model combines the aleatoric and epistemic uncertainties. Quantifying and decomposing the aleatoric and epistemic uncertainties is an active research ares in machine learning (Gawlikowski et al., 2023; Wimmer et al., 2023; Hüllermeier & Waegeman, 2021), which we leave for future work to explore for our AEA application. Similarly, the information gain based acquisition policy also requires accurate uncertainty estimation, which is challenging, especially with scarce training data on AlpacaEval and Chatbot Arena benchmarks.

### F.4. Further Analysis of the Results

From the main results in Figure 1, we found that our approach achieves exceptional estimation on AlpacaEval with near 0% absolute error, while on other benchmarks the error is obviously above 0. To better understand the factors affecting evaluation score prediction accuracy and the varying performance across different benchmarks, we conducted a detailed analysis using the HELM-Lite benchmark. We chose HELM-Lite for this analysis due to its diverse datasets covering different types of tasks and metrics. To isolate the impact of prompt selection policy, we performed this analysis using 50 randomly selected prompts as conditioning to predict the evaluation scores.

It's worth noting that prompts that are inherently difficult for language models, resulting in consistently low scores, do not necessarily translate to high prediction error for the evaluation scores. For example, if no model can solve a particular prompt, its evaluation score will always be zero, making it relatively easy to predict.

**Factors Affecting Prediction Accuracy**  For a dataset with $N$ prompts evaluated on $M$ models from the training set, with evaluation scores denoted as $\{y_{mn}\}_{m=1,n=1}^{M,N}$, we analyzed the following factors that might affect the evaluation score prediction:

- **Metric Types**: We hypothesized that discrete metrics (e.g., exact match accuracy) would be harder to predict than continuous metrics (e.g., BLEU scores).

- **Prompt Diversity**: A dataset with diverse prompts is potentially harder for LLM. The prompt diversity is measured using the average pairwise cosine similarity of prompt embeddings.

- **Task Difficulty**: We estimate the task difficulty as 1 minus the average evaluation scores across all models, i.e., $1 - \frac{1}{MN} \sum_{m,n} y_{mn}$, where higher values indicate more challenging tasks.

- **Score Diversity**: A dataset where the score distribution has a high variance can potentially lead to higher prediction error. The score diversity is calculated as the variance of evaluation scores within the dataset.

- **Task Informativeness**: We estimate the informativeness of each prompt as the variance of its evaluation scores on all models, then the task informativeness is averaged over prompts: $\frac{1}{N} \sum_n \text{Var}_m(y_{mn})$, measuring how discriminative each prompt is across different models.

- **Evaluation Variability**: Calculating the variance of the mean scores across prompts is another way to quantify the variability or diversity in the task: $\text{Var}_n(\frac{1}{M} \sum_m y_{mn})$.

**Results and Analysis**   Table F.2 shows the correlation between these factors and prediction error across 28 subsets from HELM-Lite:

*Table F.2.* Correlation between dataset characteristics and prediction error

| Factor | Spearman | Pearson |
|---|---|---|
| Metric Type | 0.783 | 0.759 |
| Prompt Diversity | -0.722 | -0.719 |
| Score Diversity | 0.216 | 0.366 |
| Task Informativeness | 0.720 | 0.781 |
| Evaluation Variability | 0.885 | 0.842 |
| Task Difficulty | -0.195 | -0.138 |

Key findings from our analysis:

1. **Metric Type Impact**: The strong correlation (0.78) confirms our hypothesis that discrete metrics are more challenging to predict than continuous ones. This helps explain why benchmarks with primarily discrete metrics (e.g., MMLU) show higher prediction errors compared to those with continuous metrics.

2. **Prompt Diversity**: Interestingly, higher prompt diversity correlates with lower prediction error (-0.72). This counter-intuitive finding suggests that while diverse prompts may be challenging for LLMs to answer, they might actually provide richer signals for predicting evaluation scores. This also aligns with the analysis that harder task for LLM does not necessarily mean harder evaluation score prediction. This is further supported by the negative correlation between Task Difficulty and prediction error.

3. **Task Informativeness**: Task Informativeness has a high correlation with prediction error, as expected, since prompts with high variance in evaluation scores are inherently harder to predict accurately.

This analysis helps explain the varying performance across different benchmarks. For instance, AlpacaEval's near-zero prediction error can be attributed to several factors:

- Its relatively small size (805 prompts) compared to other benchmarks makes dependencies easier to capture

- Its continuous win-rate scores from a logistic regression model are inherently smoother than discrete metrics

- The evaluation scores come from a single consistent source (GPT-4 evaluator)

In contrast, benchmarks like MMLU and HELM-Lite show higher prediction errors due to their larger size, predominantly discrete metrics, and more diverse evaluation criteria across different subsets.

*Table F.3.* Prediction error by metric type

| Metric Type | Prediction Error |
|---|---|
| Binary | 0.1055 |
| Real-valued | 0.0334 |

*Table F.4.* Prediction error by specific metric

| Metric | Prediction Error |
|---|---|
| BLEU-4 | 0.0302 |
| F1 Score | 0.0386 |
| Exact Match | 0.0777 |
| Quasi Exact Match | 0.0957 |
| Final Number Exact Match | 0.1134 |
| Math Equivalence Chain-of-Thought | 0.1367 |

**Impact of Metric Types**    To further investigate the impact of metric types, we analyzed prediction errors across different metrics. Tables F.3 and F.4 show detailed breakdowns:

The results clearly demonstrate that:

- Binary metrics have significantly higher prediction errors (0.1055) compared to real-valued metrics (0.0334)

- Among specific metrics, continuous measures like BLEU-4 (0.0302) and F1 Score (0.0386) show lower prediction errors

- Complex discrete metrics, particularly those involving mathematical reasoning (Math Equivalence Chain-of-Thought: 0.1367), present the highest prediction challenges

### F.5. Discussion on Random Sampling as a Baseline

Random sampling serves as a compelling baseline for efficient benchmarking due to several attractive properties:

- **Simplicity**: Implementation requires no additional models or complex selection strategies

- **Statistical Guarantees**: Supports standard statistical analysis and uncertainty quantification

- **No Cold Start**: Naturally handles new prompts without requiring historical evaluation data

- **Interpretability**: Results are easily understood and trusted by benchmark users

Our experiments show that random sampling achieves competitive performance compared to more sophisticated methods. This aligns with observations in the active learning literature, where simple random sampling often performs surprisingly well when the labeling budget is large (Lu et al., 2023; Hacohen et al., 2022; Tifrea et al., 2022). Similar phenomena have been observed in experimental design and bandit problems (Jamieson & Jain, 2018; Pacchiano et al., 2024), suggesting this is a general principle rather than specific to our setting.

However, a closer examination of our results reveals that our RL-based policy can achieve the same estimation accuracy with significantly fewer prompts:

This efficiency gain is particularly meaningful in the context of LLM evaluation, where each prompt evaluation:

- Requires significant computational resources

- May incur API costs for closed models

*Table F.5.* Comparison of prompts required to achieve similar estimation errors

| Dataset | Uniform Sampling | | RL Policy | |
|---|---|---|---|---|
| | Prompts | Error | Prompts | Error |
| AlpacaEval | 100 | 0.0051±0.0005 | 8 | 0.0051±0.0008 |
| MMLU | 100 | 0.0179±0.0010 | 35 | 0.0172±0.0023 |
| HELM-Lite | 200 | 0.0376±0.0060 | 50 | 0.0350±0.0004 |
| OpenLLM | 200 | 0.0225±0.0026 | 100 | 0.0223±0.0024 |
| MT-Bench | 40 | 0.0518±0.0127 | 23 | 0.0506±0.0118 |

- Contributes to environmental impact through energy consumption

The choice between random sampling and our method presents several trade-offs:

1. **Sample Efficiency vs. Complexity**: Our method achieves better sample efficiency but requires maintaining additional models and more complex implementation.

2. **Statistical Guarantees vs. Empirical Performance**: Random sampling provides clear statistical guarantees, while our method's theoretical properties are less well-understood despite strong empirical performance.

3. **Immediate Deployment vs. Training Requirements**: Random sampling can be deployed immediately, while our method requires initial training on historical evaluation data.

In practice, the choice between methods should be guided by specific requirements:

- When evaluation costs are high and minimizing the number of prompts is crucial, our method offers substantial benefits

- When statistical guarantees or immediate deployment are prioritized, random sampling may be more appropriate

- For large-scale evaluations where even small reductions in prompts translate to significant cost savings, the added complexity of our method may be justified

These considerations suggest that both approaches have their place in the LLM evaluation ecosystem. Random sampling serves as a reliable, interpretable baseline, while our method offers a more sophisticated alternative when maximum sample efficiency is desired. Future work could explore hybrid approaches that combine the statistical guarantees of random sampling with the efficiency gains of learned selection strategies.

### F.6. Comparison with Uncertainty Estimation Baselines

A natural alternative to our RL-based acquisition policy would be to use uncertainty estimates from the model's generations to determine which prompts to evaluate. To validate our approach, we compared our RL-based policy with two common uncertainty estimation baselines:

- **Generation Perplexity**: Using the log-likelihood of the model's generation as an uncertainty estimate

- **Semantic Entropy** (Kuhn et al., 2023): Computing entropy over semantic clusters of multiple generations using a pretrained NLI model

We conducted experiments on three recent language models (Mistral-7B, Mixtral-8x7B, and Gemma-7B) evaluating their performance on the AlpacaEval benchmark. For each model, we compare different methods for selecting prompts to evaluate, while using the same neural process model to predict scores for the remaining prompts.

As shown in Table F.6, our RL-based acquisition policy consistently outperforms both uncertainty estimation baselines across all models and selection budgets. We hypothesize that this superior performance stems from two key factors:

*Table F.6.* Comparison of acquisition policies on AlpacaEval benchmark. Values show absolute error in benchmark score estimation under different selection budgets.

| Model | Method | Selection Budget | | | |
|---|---|---|---|---|---|
| | | 10 | 20 | 50 | 100 |
| Mistral-7B | Perplexity | 0.0162 | 0.0156 | 0.0103 | 0.0088 |
| | Semantic Uncertainty | 0.0122 | 0.0175 | 0.0126 | 0.0150 |
| | RL (ours) | **0.0012** | **0.0008** | **0.0006** | **0.0006** |
| Mixtral-8x7B | Perplexity | 0.0025 | 0.0053 | 0.0058 | 0.0076 |
| | Semantic Uncertainty | 0.0095 | 0.0145 | 0.0132 | 0.0202 |
| | RL (ours) | **0.0019** | **0.0003** | **0.0003** | **0.0002** |
| Gemma-7B | Perplexity | 0.0090 | 0.0190 | 0.0136 | 0.0147 |
| | Semantic Uncertainty | 0.0148 | 0.0139 | 0.0144 | 0.0169 |
| | RL (ours) | **0.0013** | **0.0007** | **0.0009** | **0.0004** |

1. **Aligned Uncertainty Estimation**: The RL policy is trained jointly with the neural process model, allowing it to better align its uncertainty estimates with the neural process's predictive uncertainty. In contrast, inference-based uncertainty methods may capture the model's inherent uncertainty but fail to identify cases where the neural process predictions would be most uncertain or inaccurate.

2. **Sequential Decision Making**: The RL policy can adapt its selection strategy based on previously observed scores, while uncertainty-based methods make independent decisions for each prompt.

It's worth noting that these inference-dependent uncertainty methods are impractical for efficient benchmarking in practice. They require running inference on all prompts (and sometimes multiple times per prompt), which increases rather than decreases the overall evaluation cost. In contrast, our RL-based policy makes selections based solely on prompt embeddings and previously observed scores, maintaining the efficiency benefits of our approach.

## G. Computational Complexity Analysis

When comparing the computational complexity of our method to full benchmark evaluation, there are two primary components to consider: (1) the cost of acquiring evaluation scores for the selected subset of prompts, and (2) the computational overhead from running the neural process model and acquisition policies.

**Evaluation Cost Savings**    The main computational savings come from the dramatic reduction in the number of prompts requiring evaluation. Our experiments demonstrate that accurate performance estimation can be achieved with only a small fraction of the total prompts. Specifically, our method requires less than 1

**Computational Overhead**    While our method introduces some additional computation through the neural process model and acquisition policies, this overhead is minimal compared to the cost of LLM inference. The bottleneck in evaluation cost lies in LLM inference, which typically requires high-end computational resources and often involves models with billions of parameters. In contrast, our neural process model and acquisition policies are implemented with lightweight architectures (2-3 linear layers in our experiments) that can be executed efficiently on modest hardware with minimal computational resources.

In conclusion, by reducing the number of required LLM evaluations by orders of magnitude, our method offers substantial computational savings compared to full evaluation, even when accounting for the additional overhead of running our models. The lightweight nature of our approach ensures that the computational benefits scale well with increasing benchmark sizes and model complexities. This makes our method particularly valuable for large-scale benchmarks where comprehensive evaluation would be prohibitively expensive.

