# OpenReview forum: "Active Evaluation Acquisition for Efficient LLM Benchmarking"
_ICML.cc/2025/Conference — ICML 2025 poster_

### Official Review · Reviewer_GY8p · 2025-02-28

**Overall Recommendation:** 5

**Summary:**

The paper deals with efficient evaluation and offers a new way to dynamically choose which examples to use, per model. It shows great results, the improvements are very clear, the novelties are too, the writing is also mostly easy to follow.

**Claims And Evidence:**

Well supported

**Essential References Not Discussed:**

Each time I thought something was missing I instead found I was hasty to judge and it appeared somewhere else in the paper, very well documented.

**Experimental Designs Or Analyses:**

I would add something about what overheads your method adds. E.g. how much time it adds to each computation, how hard is it to implement (do you share a relevant implementation?), what other limitations may make someone not use this, as it seems really promising and the improvements are large as well.

**Methods And Evaluation Criteria:**

Well evaluated and clear. Including previous works, breaking the method to parts etc.

**Other Comments Or Suggestions:**

l 207(right) $h$ probably a typo

**Other Strengths And Weaknesses:**

The main prospect the work claims to deal with is the wrong one. The paper doesn't do anything with prompts. A certain example (abstract task, output pair) can have multiple prompts (e.g. "what is the capital of France?" or "What is the capital of france" or even "hey, dude, do you know what's the capital of France"). There is a lot of work on those things because this is an important issue in benchmarking LLMs, it took me several pages before I figured what this paper is trying to do.
An example to give the difference, you mention Polo's early work often (tinybenchmarks) but not their *prompt* related work and the name tells all the difference (Efficient multi-prompt evaluation of LLMs). It sounds like this paper should be compared to it, but in fact it is not.

**Questions For Authors:**

Why do you call your method "RL" in the graphs? Rather than the name of your method. Also your variants I would add in the legend (ours) or some other way to separate those from previous work.

**Relation To Broader Scientific Literature:**

You can cited something in l46, no need to claim it as if it was a new claim (e.g. a survey or a tutorial on efficient benchmarking or just the efficient benchmarking paper for NLP).

The distinction between active evaluation and active testing is that all active testing reduces the amound of scoring function uses and you reduce the amount of inference uses? I would make this distinction clearly (especially as you bring a new very similar name and only discuss active testing in the related work. Also, why do you add "acquisition"? Seems to imply another difference but I don't think there is. Moreover, if you claim a new approach, why not own it? Describe the difference in general and the new name and distinction and then offer your specific method.)

**Theoretical Claims:**

It is unclear from the definitions pre l76 what exactly is Y, it seems to be the score (so \in [0,1] given by some metric?) for a specific example (I assume the fact you kept using the word prompt aided the confusion). Is this argmax well defined? what is a random variable or sample in any of that? Or what is the model under which this is a probability (under a discreet set of values it is just 0/1)?
You write it in a very formal way, so one expects it is true formally, you can also explain it in a more intuitive manner or explain what your assumptions are here (probability because of uncertainty? Baseian thing with priors?) Or move things up (in l97 you define more).

---

> ### Author Rebuttal · Authors · 2025-03-29
>
> We sincerely thank the reviewer for their detailed feedback and strong endorsement of our work. We address each point below:
>
> ## Terminology: "Prompts" vs "Examples"
> We thank the reviewer for highlighting the potential confusion in our use of the term "prompt." We use "prompt" to mean benchmark examples rendered with the dedicated prompt templates, which is the exact input to the target LLM. This distinction is important because some benchmarks/datasets apply different prompt templates to the same examples - for instance, natural_questions in the HELM benchmark is evaluated in two modes with different prompt formats. This is why we focus on prompt-level evaluation efficiency rather than just example selection.
>
> We will revise the paper to clarify this terminology early on to avoid confusion with other types of prompt-related work.
>
> ## Definition of Evaluation Scores (Y)
>
> We appreciate the request for clarity regarding the definition of Y. In our formulation, Y_m represents the collection of evaluation scores for model m across all prompts in benchmark X. These scores can indeed be of mixed types - binary accuracy, continuous metrics like F1, etc. - depending on the specific dataset within the benchmark. This is explicitly mentioned in Section 2: "These scores may be of mixed types - for instance, some datasets might report binary accuracy while others use continuous metrics like F1 scores."
>
> We will improve clarity by providing a more explicit definition earlier in the paper.
> ## Probability Formulation of Eq.1
>
> We appreciate the reviewer's questions about the formal definitions in our probability formulation. The formulation in Equation (1) is indeed meant to be understood from a Bayesian perspective, and p(Y^(u)_m'|Y^(o)_m',X) represents the conditional likelihood of the unobserved scores Y^(u)_m' given the observed scores Y^(o)_m' and prompts X.
>
> Equation (1) represents an ideal case where we could directly optimize for the subset o* that maximizes the likelihood of correctly predicting unobserved scores. However, as we note in the subsequent paragraph: "the values Y^(u)_m' for a test model m' are unknown before acquisition, making direct optimization impossible." In practice, we cannot directly optimize this objective since Y^(u)_m' is unknown. Instead, we train policies that approximate this optimization based on historical evaluation data.
>
> In our revision, we will add an intuitive explanation of this formulation explaining the intuition behind our probability formulation to avoid confusion.
>
> ## Method Overhead and Implementation
>
> The reviewer raises an excellent point about discussing implementation overhead. We provide a computational complexity analysis in Appendix G showing that our method adds minimal overhead compared to the LLM inference costs it saves. Specifically:
>
> 1. The one-time training of the neural process model and acquisition policy takes 2-3 hours on a standard GPU
> 2. The policy network is lightweight, consisting of only 2-3 linear layers that execute in milliseconds
> 3. The 43-92% reduction in required evaluations vastly outweighs this overhead
>
> In our revision, we will highlight this information in the main text and clarify that we will release our implementation to help practitioners adopt our method.
>
> ## Distinction from Active Testing and Active Learning
>
> We thank the reviewer for the suggestion to clarify our positioning relative to active testing and active learning. We will revise the related work section to more clearly articulate that:
>
> 1. Active Learning typically selects examples to label for training a model
> 2. Active Testing focuses on reducing the labeling cost for evaluating model performance
> 3. Our Active Evaluation Acquisition (AEA) specifically targets reducing the number of prompt evaluations needed during LLM benchmarking
>
> We use "acquisition" to emphasize the sequential process of obtaining evaluation outcomes, but we agree this could be presented more clearly. We will revise to better own our contribution and clarify the distinctions between these related but different approaches.
>
> ## Figure Labeling
>
> We appreciate the suggestion about labeling our method in the figures. We will update all figures to clearly label our method as "AEA (ours)" or similar to distinguish it from baseline approaches.
>
> ## Typo on line 207
>
> Thank you for catching this. We will correct it in the revised version.
>
> We appreciate the reviewer's thorough reading of our work and are grateful for the strong endorsement. We believe the requested clarifications will further strengthen the paper.

---

### Official Review · Reviewer_kxi9 · 2025-03-14

**Overall Recommendation:** 3

**Summary:**

The paper presents an approach to improve the efficiency of evaluating large language models (LLMs) by selecting a subset of evaluation prompts through a learned policy. The authors claim that their RL-based approach significantly reduces computational cost while maintaining accuracy.

**Claims And Evidence:**

1.  The author assumes that evaluation scores across prompts are highly correlated, but no strong evidence is provided to support this assumption.
2. Only 5 benchmarks are selected. This could lead to misleading conclusions about the model's generalized performance.

**Essential References Not Discussed:**

N/A

**Experimental Designs Or Analyses:**

1. The experimental section lacks the statistical significance test.
2. Only 5 benchmarks are selected. This could lead to misleading conclusions about model's generalized performance.

**Methods And Evaluation Criteria:**

1. The proposed RL-based technique seems computationally expensive. Therefore, further justification is required regarding its real-world applicability.
2. Missing proof that the RL-based policy generalizes better than simpler baselines.

**Other Comments Or Suggestions:**

Would have been much better for the readers if the author demonstrates the task and the proposed methods using a figure.

**Other Strengths And Weaknesses:**

Strengths: Addresses an important problem to reduce LLM evaluation cost. Also, a well motivated paper.
Weaknesses: Only 5 benchmarks are selected. This could lead to misleading conclusions about the model's generalized performance. Also, the RL-based technique seems computationally expensive.

**Questions For Authors:**

1. Why only 5 benchmarks are selected and how do they ensure the generalized effectiveness of the proposed method?
2. Why no figure is presented to demonstrate the task and the proposed method?

**Relation To Broader Scientific Literature:**

1. Somewhat useful but adding more benchmarks and models would have been better.

**Theoretical Claims:**

N/A

---

> ### Author Rebuttal · Authors · 2025-03-29
>
> We appreciate the reviewer's feedback and concerns. Below we address the specific points raised:
>
> ## Evidence for Prompt Correlations
>
> The reviewer questions our assumption that evaluation scores across prompts are correlated. This correlation is well-documented in prior literature on LLM evaluation, including in works by Perlitz et al. (2023) and Polo et al. (2024), which we cite. Additionally, we present empirical evidence in Appendix F.4 through a detailed analysis of factors affecting prediction accuracy. This analysis shows that similar prompts (in embedding space) typically yield similar evaluation outcomes, confirming the existence of exploitable dependencies.
>
> ## Benchmark Selection and Generalizability
>
> Regarding the concern about using only 5 benchmarks, we respectfully note that our selection includes a diverse set of prominent LLM benchmarks that cover different evaluation paradigms:
> - AlpacaEval (win rate against reference models)
> - HELM-Lite (multiple tasks and metrics)
> - OpenLLM Leaderboard (multiple-choice QA)
> - MMLU (subject-specific academic tests)
> - Chatbot Arena (human preference judgments)
>
> These benchmarks represent the most widely used evaluation frameworks in the field and employ different scoring mechanisms, ensuring our method's applicability across diverse evaluation settings. Together, they comprise over 56,000 evaluation prompts across dozens of tasks and metrics, providing robust evidence for generalizability.
>
> ## Computational Efficiency of RL-based Approach
>
> The reviewer expresses concerns about the computational cost of our RL-based policy. We address this directly in Appendix G with a computational complexity analysis showing that:
>
> 1. The one-time training of our policy and neural process model is computationally inexpensive (typically 2-3 hours on a single GPU)
> 2. During inference, our policy network adds negligible computational overhead compared to the cost of running LLM inference
> 3. The savings from reducing the number of required evaluations (65-92% fewer) vastly outweigh any overhead from our method
>
> Our approach makes large-scale LLM evaluation significantly more accessible, especially when evaluating resource-intensive models or when using costly API calls for closed models.
>
> ## Statistical Significance
> Regarding statistical significance, our evaluation compares the convergence behavior of different acquisition policies across multiple runs. The standard deviations reported in our figures and detailed in Tables 1-5 provide evidence of the consistent performance advantages of our approach over baseline methods.
>
> Our results in Figure 1 clearly demonstrate that as the acquisition budget increases, our RL-based method consistently achieves lower error rates compared to baseline methods. For instance, on MMLU (Figure 1d), our method reaches an error of approximately 0.02 with just 50 prompts, while random sampling requires nearly 100 prompts to achieve the same accuracy. This pattern is consistent across all benchmarks, with our method's error decreasing more rapidly as more prompts are acquired. This faster convergence to low error rates demonstrates our method's superior efficiency in identifying the most informative prompts for evaluation.
>
> ## Visual Representation
>
> We acknowledge the reviewer's suggestion about including more visual representations. Figure 1 in our paper illustrates various benchmarks with performance plots showing acquisition budget versus absolute error. In the revised version, we will add a schematic diagram illustrating our overall approach to make the workflow more accessible to readers.
>
> We appreciate the opportunity to address these concerns and believe our work makes a significant contribution to efficient LLM evaluation, with strong empirical evidence supporting our claims.

---

### Official Review · Reviewer_NcUZ · 2025-03-15

**Overall Recommendation:** 4

**Summary:**

This paper introduces a novel RL-based method to LLM's benchmarking evaluation. From the aspects of efficiency and accuracy, they improve the accuracy in evaluation and also lower the computation overhead in the process of evaluation. They are inspired by active learning and propose their approach by modeling the dependencies across different samples. Their experimental results show their method's performance including accuracy and robustness, and also demonstrate their contribution to efficiency in LLM's evaluation field.

**Claims And Evidence:**

Yes

**Essential References Not Discussed:**

I think the necessary references are included in the method section and results part.

**Experimental Designs Or Analyses:**

Yes I did. I think their experiments are good. But there are some concerns: 1) We can see some big std in some figures and they run the code using 3 different random seeds. I think it would be better to run more times, like 5 different random seeds to lower the std of baselines.
2) Still about the absolute error, the paper did not write why the metric is good. Because the numbers of all results are so small, I can't conclude how powerful their method is. Maybe they can include some instructions on why they use the metric.

**Methods And Evaluation Criteria:**

The methods make sense. Because the active learning can truly improve the efficiency and keep the accuracy. They include the active learning and RL-based method to model the dependencies to achieve their goals. Also their method is based on the neural process and we can find the relevant literature to support the parts in method. And the method is consistent with their motivation.

For their evaluation criteria, I think it's good. For absolute error, it can show method's effectiveness. But I am not sure whether it's the best. Or maybe it's not enough. For example, they could include the running time to prove they improve the efficiency. And from the numbers of  absolute error, different results are not very distinguishable. Is there any other better criteria. Or maybe they could show the metric they use is most used often.

**Other Comments Or Suggestions:**

I don't have other comments.

**Other Strengths And Weaknesses:**

Other Strengths: They present their paper in a clear way to show the pipeline and motivation along with corresponding supporting results. Their algorithms'  pseudo and figures in paper are clear and great.
Other Weaknesses: If they could introduce the neural process in the preliminary more, it would be better.

**Questions For Authors:**

Could you run more different random seeds to decrease the variance of baselines? That would show your method's performance better. Current some of figures are not so clear due to big shaded areas caused by variance.

**Relation To Broader Scientific Literature:**

They introduced active learning to LLM's evaluation. I think their work can inspire later related research areas. In the future, there may be more works to lower the computation cost in LLM's evaluation and keep the performance at the same time. Also in the other fields related to LLM, how to improve the accuracy of other LLM's tasks and also improve the efficiency.

**Theoretical Claims:**

I think their theoretical parts are correct. They didn't put any theorems/lemmas. Just from the equations in paper, I think they are clear and not wrong. For example, from the equation7-9, they are clear to define a reward way in RL to show their novel ideas in method. So far, I didn't see any wrong things in theoretical parts.

---

> ### Author Rebuttal · Authors · 2025-03-29
>
> We sincerely thank the reviewer for their thorough evaluation of our work and the positive recommendation. Below, we address the specific concerns and questions raised:
>
> ## Evaluation Metrics and Result Interpretation
>
> The reviewer raised questions about our use of absolute error as an evaluation metric and the distinguishability of results. We chose absolute error because it directly measures how accurately our method estimates the true benchmark performance, which is the primary goal of our work.
>
> Regarding the distinguishability of results: This is due to the fact that given enough acquisition budget, all selection methods can eventually approximate the overall performance. However, as clearly demonstrated in our figures, our RL-based approach decreases the error much more quickly as the acquisition budget increases, which demonstrates its superiority. In Appendix F.5, we conduct a detailed analysis comparing our approach to random sampling, showing that we can achieve the same level of error using 35-92% fewer prompts across different benchmarks. This substantial reduction in required evaluations represents significant cost savings in practical applications.
>
> For instance, on MMLU, our method achieves the same accuracy with just 35 prompts as random sampling does with 100 prompts - a 65% reduction that would translate to proportional cost savings when evaluating new models.
>
> ## Statistical Significance and Random Seeds
>
> We conducted experiments with three random seeds to keep comparison consistent across all methods. The larger variance observed in baseline methods actually further demonstrates the superiority of our approach, which maintains more consistent performance across different initialization seeds. For better result representation, we have included detailed tables in the appendix (Table F.1) that show performance statistics across all runs.
>
> ## Neural Process Background
> We appreciate the suggestion to provide more background on neural processes. In the revised version, we will expand Section 3.1 to include:
> 1. A more intuitive explanation of how neural processes model stochastic functions
> 2. A brief comparison with related approaches like Gaussian Processes
> 3. A clearer explanation of why neural processes are particularly well-suited for capturing dependencies across evaluation prompts
>
> This additional background will make our methodology more accessible to readers who may be less familiar with these techniques.
>
> We appreciate the reviewer's positive assessment of our paper's organization, algorithms, and figures. We're committed to further improving the clarity and impact of our contribution in the final version.

---

### Official Review · Reviewer_srP5 · 2025-03-19

**Overall Recommendation:** 4

**Summary:**

The paper focuses on LLM efficient evaluation , that is, estimating overall performance based on a subset of data. The authors first model dependencies across evaluation prompts using neural processes, then analyze various selection methods and propose a RL-based method.  Additionally, for the cold start problem, the authors propose a semi-supervised approach to solve it. Finally, they compare various methods on five benchmarks and conduct a comprehensive analysis.

**Claims And Evidence:**

Yes, the experiment results support their claims well.

**Essential References Not Discussed:**

You should cite the "Anchor Points: Benchmarking Models with Much Fewer Examples" published in ACL 2024, although you have discussed their method: Clustering

**Experimental Designs Or Analyses:**

Yes. The baselines selection in section 3.2 is comprehensive, including random policy, clustering, IRT, uncertainty sampling, and so on.

**Methods And Evaluation Criteria:**

Yes, the methods, baselines and the selected benchmark are appropriate.

**Other Comments Or Suggestions:**

A suggestion is that you should highlight the practical applications of efficient evaluation in the introduction, for example, it could be used for rough testing during the model development process, etc.

**Other Strengths And Weaknesses:**

Strengths:
- The experiments and analyses were both conducted comprehensively.
- The paper is well-structured.

Weaknesses:
- Although RL-based methods have achieved better performance than random sampling, it is more complex as they require evaluation data history, training a VAE model, and training an RL model. By comparison, the random sampling is still a strong baseline. More importantly, it is simple and efficient. So, whether the RL-based method is more suitable for practical applications remains uncertain.

**Questions For Authors:**

Please explain how to evaluate a new model using your method from scratch, given a benchmark that already contains test results for some models. Specifically, outline each step involved and estimate the workload required, such as training models. Additionally, discuss whether, given this workload, researchers would be more inclined to adopt your method or simply use random sampling.

**Relation To Broader Scientific Literature:**

The work focus on efficient evaluation of LLM. Most important related work have been discussed in this paper, such as CAT and Clustering- IRT

**Theoretical Claims:**

Yes. The reasoning process of ELBO in section 3.1 is correct.

---

> ### Author Rebuttal · Authors · 2025-03-29
>
> We sincerely thank the reviewer for their thorough evaluation of our work and the positive recommendation. Below, we address the points raised:
>
> ## Practical Applications of Efficient Evaluation
>
> We appreciate the suggestion to highlight practical applications of efficient evaluation in the introduction. In the revised version, we will emphasize how our method can be particularly valuable during model development for:
>
> 1. Enabling more frequent intermediate evaluations during training, allowing earlier detection of issues
> 2. Supporting more extensive hyperparameter tuning by reducing evaluation costs per configuration
> 3. Facilitating rapid comparisons between model variants during research iterations
> 4. Reducing API costs for closed-source models during the development phase
> 5. Decreasing environmental impact through reduced compute requirements
>
> ## Implementation Workflow for New Models
>
> Regarding how to evaluate a new model from scratch, we thank the reviewer for prompting us to better highlight our existing detailed workflow in Appendix A, which provides a comprehensive description of the training and deployment procedure. As described there, our approach involves:
>
> 1. **One-time setup**: Training the neural process model and acquisition policy on historical benchmark data
> 2. **Per-model evaluation**: Sequentially selecting prompts, obtaining scores, and predicting remaining scores
>
> Once this setup is complete, the evaluation process for each new model adds minimal computational overhead compared to the actual LLM inference costs.
>
> ## Complexity vs. Random Sampling Trade-off
>
> We acknowledge the valid concern about complexity versus simplicity. While random sampling is indeed simpler, our results demonstrate substantial efficiency improvements (43-92% fewer evaluations) that justify the additional setup complexity in many scenarios. As detailed in Appendix F.5, our approach offers particularly compelling benefits for:
>
> - Organizations conducting ongoing LLM development and benchmarking
> - Evaluations of large or expensive models where each prompt evaluation is costly
> - Scenarios requiring adaptation to new model families or previously unseen prompts
>
> The choice between approaches depends on the specific evaluation context, with our method becoming increasingly beneficial as evaluation costs or frequency increase. We'll ensure these considerations are more prominently highlighted in the main text of our revision.

---

> > ### Comment · Reviewer_srP5 · 2025-04-04
> >
> > Thank you for your detailed response and I will maintain my positive score.

---

### Official Review · Reviewer_wFjY · 2025-03-26

**Overall Recommendation:** 3

**Summary:**

This paper proposes a large language model (LLM) evaluation method, which considers dependency modeling and subset selection to improve efficiency. The authors develop a model that captures dependencies across evaluation prompts and propose subset selection policies based on these dependencies. Extensive experiments on multiple LLM evaluation benchmarks demonstrate the superiority
of the proposed method in providing accurate performance estimation with minimal acquisition budget.

## update after rebuttal

The authors' rebuttal solves most of my concerns. I regard this paper as a weak accept case and maintain my score.

**Claims And Evidence:**

The claims made in the submission are supported by clear theoretical and empirical evidence.

**Essential References Not Discussed:**

There are some important papers about active evaluation of natural language generation (NLG), which should be properly discussed in this paper. For example, [1] investigates active learning in the related evaluation setting (i.e., pairwise comparison).

[1] Active Evaluation: Efficient NLG Evaluation with Few Pairwise Comparisons. ACL 2022

**Experimental Designs Or Analyses:**

The experimental designs or analyses are almost sound.

**Methods And Evaluation Criteria:**

The proposed method and evaluation criteria almost make sense for the problem. However, there is a potential risk that the selected subset for different groups of LLMs may vary. Thus, the comparison result of two LLMs' performance may be affected by the group of LLMs in the training dataset, which seems unnatural.

**Other Comments Or Suggestions:**

None.

**Other Strengths And Weaknesses:**

Strengths:

1. The research problem about efficient LLM evaluation is interesting and realistic, because the cost of LLM evaluation increases quickly since the model parameters become larger.
2. The proposed method is sound and convincing with dependency modeling and subset selection.
3. This paper is well-organized and overall easy to follow.

Weaknesses:

1. I wonder whether the proposed method can effectively adapt to new models and benchmarks, since the core challenge of LLM evaluation compared with traditional NLG tasks' evaluation is the high requirement of generalization ability. The authors should add more theoretical or empirical analysis about this point.

**Questions For Authors:**

I have included my questions in other parts of the review.

**Relation To Broader Scientific Literature:**

Compared with the works in the broader scientific literature, the key contributions of this paper is to improve LLM evaluation efficiency via dependency modeling and subset selection.

**Theoretical Claims:**

I have checked most of the theoretical part in this paper and found no obvious errors.

---

> ### Author Rebuttal · Authors · 2025-03-29
>
> We sincerely thank the reviewer for their thoughtful assessment of our work. Below, we address the key points raised:
>
> ## Concern about Subset Selection and Fairness Across Models
>
> The reviewer raises a valid concern about whether variations in selected subsets for different groups of LLMs might affect fair comparison between models. This is indeed an important consideration that we would like to clarify:
>
> 1. Our framework maintains fair comparability by using both the acquired scores on selected prompts AND predicted scores on the remaining prompts for final benchmark estimation. This ensures that all models are evaluated on the full benchmark (either via direct evaluation or accurate prediction), enabling fair comparison regardless of which specific prompts were selected for each model.
>
> 2. As demonstrated in Table 2 of our paper, our prediction-based estimation approach ("w/ pred") consistently outperforms direct aggregation of only the acquired evaluation scores ("w/o pred"). This confirms that our method maintains benchmark integrity while improving efficiency.
>
> 3. We conducted a specific experiment examining model bias (Section 5, Fig. 2) where we deliberately evaluated on models from different families than those used for training. The results demonstrate that our method remains effective even in this challenging scenario, indicating robustness to the training dataset composition.
>
> ## Adaptation to New Models and Benchmarks
>
> The reviewer asks about our method's ability to adapt to new models and benchmarks, which is indeed crucial for LLM evaluation:
>
> 1. For new models: Our approach is explicitly designed for evaluating new, unseen models. The train-test splits in our experiments (particularly for Open LLM Leaderboard where we use chronological splitting) ensure that our method is validated on truly new models not seen during training. The consistently strong results across various benchmarks demonstrate the generalization capability to new models.
>
> 2. For new prompts (cold start problem): We explicitly address this in Section 5 with a dedicated experiment (Fig. 3) where 15 MMLU subsets are treated as "cold start prompts" with no historical scores. Our RL-based policy successfully generalizes to these completely new prompts due to our policy architecture that explicitly incorporates prompt representations.
>
> 3. For adapting to distribution shifts: We could further enhance adaptation through continual learning approaches where the neural process model and acquisition policies are jointly updated as new models are evaluated - a promising direction for future work that we mention in the conclusion.
>
> ## Missing Reference on Active Evaluation
>
> We thank the reviewer for highlighting the paper "Active Evaluation: Efficient NLG Evaluation with Few Pairwise Comparisons". This is indeed a relevant work that should be discussed in our paper. This paper focuses on efficiently identifying top-ranked NLG systems using pairwise comparisons, specifically applying dueling bandit algorithms to reduce the number of human annotations required.
>
> Our work shares the goal of improving evaluation efficiency but addresses a complementary problem: reducing the number of prompts needed to benchmark individual LLM performance rather than ranking systems through pairwise comparisons. We will incorporate a thorough discussion of this paper in our revised manuscript, acknowledging their valuable contributions to efficient evaluation methods and highlighting how both approaches serve the broader goal of making NLG/LLM evaluation more practical and accessible.
>
> ## Additional Comments
>
> We note the reviewer's positive assessment of our paper's strengths, including:
> - The importance of the research problem
> - The soundness of our proposed methods
> - The clarity and organization of the paper
>
> These align with our goal of addressing a practical challenge in LLM development while maintaining scientific rigor.
>
> ## Conclusion
>
> We appreciate the reviewer's recommendation and constructive feedback. We believe our work makes a significant contribution by dramatically reducing evaluation costs (by 43-92% across benchmarks) while maintaining accurate performance estimation. This enables more efficient and sustainable LLM development, especially for researchers with limited computational resources.
>
> In the revised version, we will:
> 1. Further clarify how our approach maintains fair comparability between models
> 2. Incorporate the suggested reference and discuss its relevance
> 3. Expand our discussion on adaptation to new models and benchmarks
>
> Thank you for the opportunity to address these points and strengthen our paper.

---

### Decision · Program_Chairs · 2025-05-01

**Decision:**

Accept (poster)

**Comment:**

This paper studies efficient evaluation of large language models, and proposes a novel RL-based policy for selecting a subset of test examples by leveraging captured dependencies across them. Empirical results demonstrate that the proposed approach can substantially reduce the number of test examples needed while maintaining accurate performance estimation. Reviewers unanimously recommend acceptance.